# Mechanically transduced immunosorbent assay to measure protein-protein interactions

Christopher J Petell[1,2], Kathyrn Randene[3], Michael Pappas[4], Diego Sandoval[4], Brian D Strahl[1,2], Joseph S Harrison[3]*, Joshua P Steimel[5]*

[1]Department of Biochemistry and Biophysics, The University of North Carolina School of Medicine, Chapel Hill, United States; [2]UNC Lineberger Comprehensive Cancer Center, University of North Carolina, Chapel Hill, United States; [3]Department of Chemistry, University of the Pacific, Stockton, United States; [4]Department of Biological Engineering, University of the Pacific, Stockton, United States; [5]Department of Mechanical Engineering, University of the Pacific, Stockton, United States

**Abstract** Measuring protein-protein interaction (PPI) affinities is fundamental to biochemistry. Yet, conventional methods rely upon the law of mass action and cannot measure many PPIs due to a scarcity of reagents and limitations in the measurable affinity ranges. Here, we present a novel technique that leverages the fundamental concept of friction to produce a mechanical signal that correlates to binding potential. The mechanically transduced immunosorbent (METRIS) assay utilizes rolling magnetic probes to measure PPI interaction affinities. METRIS measures the translational displacement of protein-coated particles on a protein-functionalized substrate. The translational displacement scales with the effective friction induced by a PPI, thus producing a mechanical signal when a binding event occurs. The METRIS assay uses as little as 20 pmols of reagents to measure a wide range of affinities while exhibiting a high resolution and sensitivity. We use METRIS to measure several PPIs that were previously inaccessible using traditional methods, providing new insights into epigenetic recognition.

*For correspondence:
joseph.scott.harrison@gmail.com
(JSH);
jsteimel@pacific.edu (JPS)

Competing interest: See
page 17

Reviewing editor: Raymond E
Goldstein, University of
Cambridge, United Kingdom

## Introduction

Protein-protein interactions (PPIs) are essential to cellular biology and both high- and low-affinity interactions are required to maintain robust and dynamic responses in biological circuits (*Nooren and Thornton, 2003*; *Kastritis and Bonvin, 2013*; *Evans, 2001*). Low-affinity interactions are commonly leveraged, as is seen for multivalent recognition (*Markin et al., 2010*), readers of highly abundant proteins, and in protein allostery (*Daily and Gray, 2009*). In particular, recognition of the epigenome is recognized to rely on the interplay between post-translational modifications (PTMs), like methylation, phosphorylation, and ubiquitination (*Yau et al., 2017*; *McGinty and Tan, 2016*; *Patel, 2016*). Furthermore, multidomain-containing proteins are often regulated by allostery through weak interdomain interactions (*Gladkova et al., 2018*; *Peng, 2015*). Increasingly, the importance of weak interactions or relatively small changes in PPI affinity has been realized.

Despite the increasing sophistication of studying PPIs, biochemical characterization of these weaker and similar strength interactions remain a significant hurdle. Many techniques are useful for examining protein-binding strength, each with its own set of limitations (*Rowe, 2011*; *Syafrizayanti et al., 2014*; *Pollard, 2010*). However, virtually all the commonly used techniques to measure biological interactions, for example, like ELISA, FP, SPR, NMR, BLI, AUC, and ITC, rely on the law of mass action, and to measure protein binding affinities in the µM range and above, highly

concentrated proteins or ligands are required (*Xing et al., 2016*). For many systems, obtaining such large quantities of materials can be unattainable. Furthermore, high protein concentrations leads to thermodynamic non-ideality and proteins can aggregate, self-associate, and non-specific interactions occur, thus obfuscating the binding signal (*White et al., 2010*; *Saluja and Kalonia, 2008*). NMR is the gold standard method to measure weak interactions; however, in addition to requiring copious amounts of materials, the proteins must also be isotopically labeled, a single affinity measurement requires substantial instrument time and complex data analysis, and of all the methods mentioned is the lowest throughput. BLI and SPR are methods that can measure interactions while using a small quantity of the immobilized partner, however, binding is still governed by mass action and the soluble analyte must be at concentrations above the $K_d$; for weak binders, this can still use large quantities of materials (*Helmerhorst et al., 2012*; *Weeramange et al., 2020*). Additionally, the signal is highly dependent on the mass change of the interaction, and for smaller ligands, binding to larger molecules this signal could be small. Moreover, discerning between background binding and specific binding can be difficult, especially for weak interactions which requires the analyte to be at a high concentration.

Another difficulty in determining the binding affinity of PPIs arises when measuring similar strength interactions, for example, two- to fivefold differences. Several factors contribute to this limitation, but determining the active fraction of protein is significant because, for most fitting techniques, the calculated affinity is a dependent variable of the protein concentration (*Jarmoskaite et al., 2020*; *Hulme and Trevethick, 2010*). Moreover, to achieve accurate fitting of a protein binding isotherm requires accurate determination of the end point of the saturation curve, which for weak interactions necessitates high concentrations of ligand. Another factor in differentiating similar strength interactions is that most binding measurements have low statistical power due to the resource intensiveness of performing multiple replicates. A method where binding strength can be measured independent of protein concentration, that uses small amounts of reagents, and that has high statistical power would be valuable.

Here, we present a novel approach to measuring the strength of biological interactions that is moderately high-throughput, requires a minimal amount of protein material, and can measure a wide range of $K_d$ values from $10^{-2}$ to $10^{-15}$ M. This technique was initially inspired by the rolling of biological cells, like neutrophils exhibiting haptotaxis on endothelial cells. Neutrophil motion is driven by chemical or ligand gradients (*Voisin and Nourshargh, 2013*). The neutrophils roll on the endothelial cells due to PPIs between the cell surface receptors. The PPIs increase the effective friction between the two cells, allowing the rotational motion to be converted into translational displacement. We aimed to create a single particle biomimetic technique that leveraged this fundamental physical concept of friction to produce a mechanical signal to indicate binding events, the Mechanically Transduced Immunosorbent assay (METRIS). METRIS utilizes protein functionalized ferromagnetic particles to mimic the rolling cells. These ferromagnetic particles are made active via actuation of an externally applied rotating magnetic field and the particles proceed to roll, henceforth referred to as rollers, and translate across the surface using a similar mode of locomotion as the neutrophils. When the rollers are placed on a functionalized surface, the amount of rotational motion converted into translational motion depends on the effective friction between the rollers and the substrate. That effective friction scales with the strength of the binding interaction. Thus, a higher affinity PPI between the roller and the substrate will result in a larger translational displacement of the roller. Since both the roller and surface have immobilized proteins, the method is not dependent on mass action and requires approximately 20 pmols to measure PPIs regardless of their strength.

Using the METRIS assay, we reproduced well-characterized binding preferences for two different methyllysine histone reader domains (*Gatchalian et al., 2013*; *Kuo et al., 2012*) and weak interactions between the E2 Ube2D (*Buetow et al., 2015*) and UBL-domains (*DaRosa et al., 2018*). These affinities range between $10^{-4}$ and $10^{-6}$ M. However, we were also able to measure several weaker interactions between unmodified histone peptides, which allowed us to measure the $\Delta\Delta$Gs for the phospho/methyl switch phenomenon in DIDO1-PHD (*Andrews et al., 2016*). Finally, we also show that this method can be used to measure a weak interdomain interaction between the isolated UHRF1-UBL domain and SRA domain, which is known to control the E3 ligase specificity and epigenetic DNA methylation inheritance (*Foster et al., 2018*; *DaRosa et al., 2018*). Collectively, our

results show that the METRIS assay can be a very powerful technique which has the potential to provide additional insight into PPI interactions that were difficult to measure using other methods.

## Results

### Rolling parameter scales with interaction affinity of PPI

In the METRIS assay, rollers are placed in a Helmholtz coil inspired apparatus (see *Figure 1A* and *Figure 1—figure supplement 1*) where an externally rotating magnetic field is applied at a constant frequency, $\omega$. The permanent magnetic moment of the roller, couples with the applied magnetic field, producing a magnetic torque and subsequent rotation of the ferromagnetic bead (*Steimel et al., 2014*; *Sing et al., 2015*). In the absence of effective friction, the rollers would rotate mostly in place with the frequency of the applied magnetic field; however, effective friction induced by binding between the rollers and the substrate will convert some of that rotational motion into translational displacement, $\Delta x$, thus indirectly measuring the effective friction between the substrate and the rollers. Since the magnetic field is many orders of magnitude stronger than the strength of noncovalent interactions, in this system the higher the effective friction between the rollers and the surface corresponds to larger translational displacement. The effective friction is determined by the strength and density of PPIs between the roller and the coated substrate. Thus, the translational displacement will scale with the density and affinity of the PPIs being measured, such that a higher $\Delta x$ corresponds to a higher affinity. However, the displacement is also a function of several other parameters, specifically the diameter of the roller, $D$, and the frequency of rotation of the applied magnetic field, $\omega$. Here we define a dimensionless parameter, which is the ratio of the observed translational displacement of the roller to the maximum theoretical translational displacement of a sphere that we refer to as the rolling parameter, RP (*Figure 1B*)

$$RP = \frac{\Delta x}{\pi D \tau \omega} \tag{1}$$

where $\Delta$x is the translational displacement of the roller, D is the diameter of the roller, $\tau$ is the actuation period of the magnetic field, and $\omega$ is the rotational frequency of the magnetic field. The RP is a dimensionless parameter that varies from 0 to 1 where 0 is no translational displacement and one being a sphere perfectly rolling at a single hinge point and translating a distance equivalent to its circumference. Here, the sphere also undergoes a number of rotations given by product of the rotational frequency of the magnetic field and actuation time. The density of the interactions between the roller and the substrate are kept as constant as possible from experiment to experiment by fully saturating both the rollers and the substrate with proteins and peptides. As described in the Materials and methods, both the rollers and substrate are coated $50\times$ the theoretical number of binding sites, so virtually all the sites should be occupied. Additionally, a series of washing steps are carried out to make sure no unbound protein or peptide remains on the surface. If the surface was not uniformly functionalized, the roller's displacement in these regions would be detected by correlations to either the individual roller or areas on the substrate. However, no such anomalies were observed in these experiments.

To measure the translation displacement $\Delta x$ and to calculate the RP of the rollers, a clockwise (CW) field was actuated at $\omega = 1 Hz$ for $\tau = 5$ s. The field was then turned off for $\tau = 5$ s. A counterclockwise (CCW) field was actuated at $\omega = 1 Hz$ for $\tau = 5$ s and then the field was turned off for $\tau = 5$ s again. This process was repeated 18 times, and several example images of rollers and roller trajectories can be seen in *Figure 1C* and *Figure 1D* and in the supplemental videos. Appropriate parameters for the magnetic field strength and frequency were previously determined (*Steimel, 2017*). The rolling parameter is calculated from the observed roller translational displacement divided by the maximum theoretical translational displacement of a rolling sphere where all the rotational torque is converted into translation, so the rolling parameter varies from 0 to 1. A rolling parameter of 0 corresponds to a surface with no effective friction. Experimentally, a rolling parameter of 0 is never observed due to hydrodynamic friction between the roller and the substrate. A rolling parameter of 1 corresponds to the maximum theoretical rolling of a sphere.

We first measured the rolling of streptavidin rollers on an avidin surface or a biotin surface. Still images of a CW (top) and CCW (bottom) actuation (*Figure 1C*) show that on the streptavidin surface

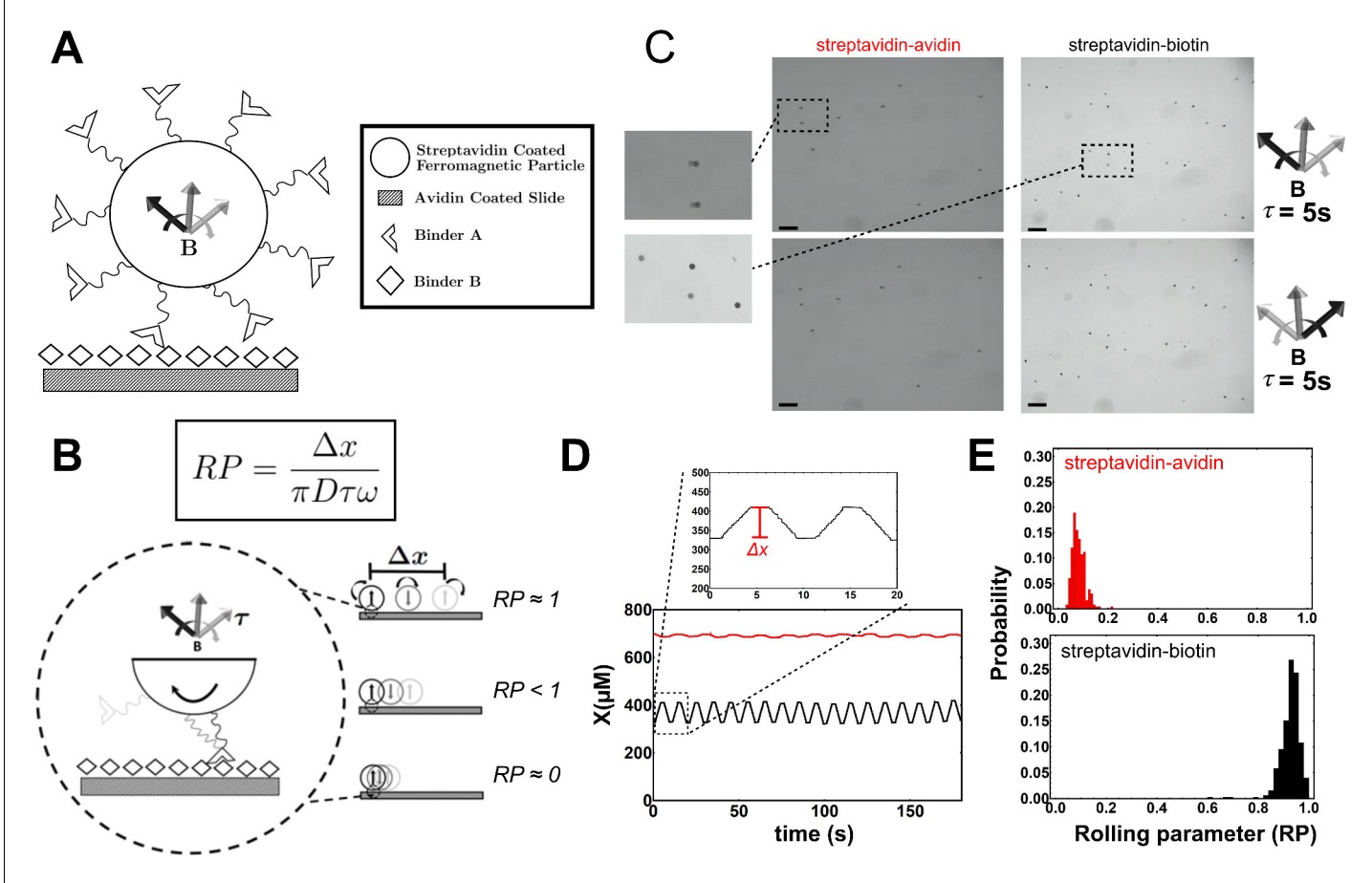

**Figure 1.** Experimental schematic of the Mechanically Transduced Immunosorbent Assay (METRIS) used to measure protein-protein interactions. (**A**) General schematic of roller and surface functionalization. Both binder A and binder B are attached to the roller or the surface by biotin-streptavidin interactions. The direction of the rotating magnetic field is indicated by the curved arrow. (**B**) The rolling parameter (RP) is a dimensionless parameter that measures rolling. The RP is calculated by taking the ratio between the observed displacement of a roller, $\Delta x$, and the maximum theoretic rolling of a sphere, which is calculated from the circumference of each spherical particle $\Pi D$, the frequency of the rotation of the magnetic field $\omega$, and the actuation time $\tau$. In the schematic three scenarios are depicted: (top) a RP of 1 where the roller moves the maximum theoretical displacement, (middle) a RP less than 1, and (bottom) a RP near 0 where the particle does not move. (**C**) Representative microscopy images of streptavidin rollers (black points) on an avidin (left) or a biotin surface (right). The scale bar in black is 100 µm and the images size are 1.28 mm × 0.96 mm. The position of the rollers prior to magnetic field actuation are indicated by the transparent spots and after actuation is opaque. The top panels are after a CW actuation and the bottom is after a CCW actuation. The magnification illustrates the difference between the null (streptavidin-avidin) and biotin-streptavidin interaction translational displacement. (**D**) Plot of a single roller from a streptavidin-biotin (black) and an avidin-streptavidin (red) experiment. The y-axis (**X**) represents the position of each roller in the field of view. The magnification above shows how $\Delta x$ is calculated for each roller by subtracting the preactuation position from the postactuation position. CW and CCW actuations are repeated as described in the methods. Translational displacement is calculated as a vector. (**E**) The distribution of rolling parameters (RPs) from the streptavidin-avidin (N=8 rollers) and biotin-streptavidin (N=9 rollers) experiments. RP is calculated using the equation in (**B**) for each actuation period for each roller so the distributions contain N × 36 points. See *Figure 1—source data 1* for the rolling parameter for each actuation.

The online version of this article includes the following video, source data, and figure supplement(s) for figure 1:

**Source data 1.** Rolling parameter from all rolls for either biotin-streptavidin and avidin-streptavidin.

**Figure supplement 1.** METRIS apparatus.

**Figure 1—video 1.** Experimental video of streptavidin-coated rollers on an avidin-coated substrate and sped up 8X.
https://elifesciences.org/articles/67525#fig1video1

**Figure 1—video 2.** Experimental video of streptavidin-coated rollers on a biotin-coated substrate sped up 8X.
https://elifesciences.org/articles/67525#fig1video2

the rollers hardly move, while on the biotin surface the rollers translate well over 100 μm. A full trajectory for a roller on the biotin and streptavidin surfaces (*Figure 1D*) show the $\Delta x$ for each roller remains relatively constant through each actuation (see *Figure 1—video 1* and *Figure 1—video 2* for movies of the experiment). $\Delta x$ is calculated for each actuation for each roller and then converted to a rolling parameter (RP) (*Figure 1D*). The RPs have a gaussian distribution (*Figure 1E*) and the average RP on the avidin surface is 0.081 ± 0.004 while on biotin we observed a RP of 0.918 ± 0.002. The interaction between biotin and streptavidin is reported to be $K_d = 10^{-15}$ M. While it is impossible to know the true $K_d$ value for a null interaction, the weakest PPI measured are in the $10^{-2}$ M range (*Yoo et al., 2016*) and enzymes with $K_d$ values in the $10^{\circ}$ M range have been reported (*Bar-Even et al., 2011*), so we assume that null interaction must be between $10^{\circ}$ M and the concentration of water $5.5 \times 10^2$ M, and we settled on $10^{\circ}$ M as an estimation of the null interaction, which based on our subsequent fitting seems like a suitable assumption. These two values provide an idea about the range of affinities that can be measured with METRIS.

## DIDO1-PHD phospho/methyl switch characterized by METRIS

Next, we wanted to determine whether we can quantitatively correlate the measured RP to binding affinities for known PPIs and test the robustness of the METRIS assay as an experimental approach to measure PPIs. We focused our attention on weak interactions and interactions between several protein pairs that are similar in binding strength, given that these PPIs are typically difficult to accurately measure. We first examined the well-established interaction between DIDO1-PHD and H3K4 methylation. DIDO1 is responsible for interchanging between active and silent chromatin states in embryonic stem cells, and its chromatin localization is regulated through a phospho/methyl switch, where phosphorylation of H3T3 evicts DIDO1 from chromatin during mitosis (*Fütterer et al., 2017*; *Di Lorenzo and Bedford, 2011*; *Liu et al., 2014*). The affinities for mono-, di-, and trimethylated peptides are well described in the literature (*Gatchalian et al., 2013*) and interactions with the unmodified peptide and H3T3pK4me3 were too weak to be measured in the experiment setup. H3K4 peptides and DIDO1-PHD were both immobilized to the rollers and substrate through biotin-streptavidin interactions. The H3 N-terminus (a.a. 1–20) was biotinylated and immobilized on the roller, and biotinylated avi-tagged GST-DIDO1-PHD was attached to the substrate. DIDO1 has a preference for H3K4me3 > H3K4me2 > H3K4me1 (*Gatchalian et al., 2013*). The measured $\Delta x$ and RP match this preference, with the largest rolling parameter for H3K4me3 (0.233 ± 0.012) > H3K4me2 (0.213 ± 0.010) > H3K4me1 (0.176 ± 0.005) and H3 and H3T3pK4me3 being the lowest, although still above the baseline rolling parameter value of 0.081 (*Figure 2A and B*). While the overall change to the RPs is small, these differences are all statistically significant because the data set has good statistical power and small percentage errors (<5%) (*Figure 2—source data 2*). Additionally, the distribution of rolling parameters can be found in *Figure 2—figure supplement 1A*, *Figure 2—video 1* and *2*, and *Figure 2—animation 3*, show the rolling for this family of interactions.

In order to correlate binding affinity to RP, we developed an empirical fitting method based on available data. We noticed that the log-log plot of $K_d$ vs. RP showed a linear relationship between the three known DIDO1-PHD binding interactions to the methylated peptides ($R^2 = 0.995$). We also included a no-binding avidin-streptavidin interaction (RP=0.081) estimated to have a $K_d = 1$ M and the streptavidin-biotin interaction where $K_d = 10^{-15}$ M (*DeChancie and Houk, 2007*; *Figure 2C*). Overall, this experiment suggests that there is a linear dependence of the log of the RP to the log of $K_d$ that spans roughly fifteen-orders of magnitude.

There is a clear correlation between RP and the measured $K_d$, the equilibrium constant for interactions, despite METRIS being a non-equilibrium technique. $K_d$ is a ratio between the first-order dissociation rate ($K_{off}$) and the second-order association rate ($K_{on}$) (*Sanders, 2004*). For most PPIs, the $K_{on}$ rates are very similar, and thus the $K_d$ constant is mostly dependent on $K_{off}$. However, kinetic constants for binding interactions are rarely reported since few techniques can access this information, so for many interactions, only $K_d$ is known. Since we do not have a theoretical model that relates RP to $K_d$, we sought to use an empirical fitting method based on the excellent correlation we observed between RP and $K_d$ (*Figure 2C*). Using this fitting method, we could reproduce the literature $K_d$ values with high accuracy; all of the predicted $K_d$ values were roughly twofold tighter than the published values (*Gatchalian et al., 2013*) and the fold difference between the different methylation states is similar (*Figure 2D*). Remarkably, we were also able to estimate METRIS-$K_d$ values for the weak interaction between the H3T3pK4me3 peptide (340 μM ± 90) and the unmodified H3 tail

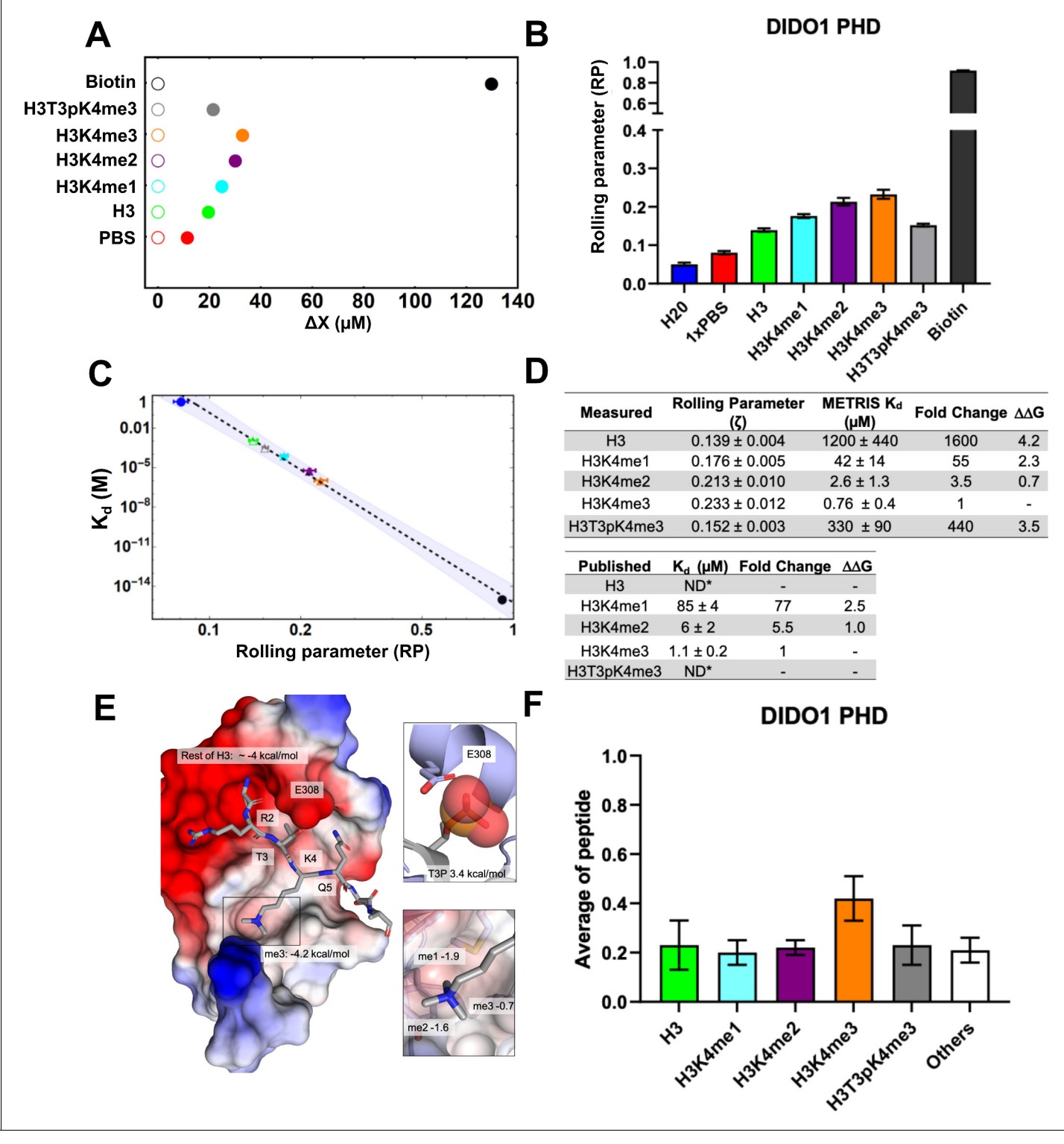

**Figure 2.** DIDO1-PHD interactions with H3 peptides characterized using METRIS. (A) Plot showing the average translational displacement per actuation for the rollers coated with the indicated H3K4 methylated peptide on a DIDO1-PHD surface. Streptavidin-biotin and streptavidin-avidin (PBS) are included for references. See *Figure 2—source data 2* for results of statistical analysis; all comparisons are statistically significant (p<0.0001). (B) Plot showing the calculated average rolling parameter per interaction. (C) Log-Log plot of the rolling parameters (RP) from panel B with the reported $K_d$s. Extrapolated points for the unknown interactions are represented by unfilled markers, and the 95% confident interval for the fitting is depicted. (D) Table of rolling parameters and associated $K_d$ estimates for the DIDO1-PHD interactions. Fold change is calculated as the ratio between the $K_d$ values for the indicated peptide and for H3K4me3. These ratios are used to calculate $\Delta\Delta G$ at T=298K. The published values are from *Gatchalian et al., 2013*

*Figure 2 continued on next page*

*Figure 2 continued*

using NMR (me1) and tryptophan fluorescence (me2/3); *ND = Not determined. (**E**) Image of the DIDO1-PHD crystal structure with H3K4me3 peptide, with the PHD surface electrostatic potentials shown (red = negative, blue = positive), the ΔΔG for K4me3, and the estimated ΔΔG for the rest of the peptide. The PTM reader sites are shown with greater detail to the right. Here, ΔΔG is calculated between the sequential methyl states, and the ratio of H3T3pK4me3 and H3K4me3 give the ΔΔG for T3p. (**F**) Results of the DIDO1-PHD histone peptide microarray assay against the indicated peptides (see *Figure 1—figure supplement 1B* for results of all peptides). Only H3K4me3 is statistically significant (P<0.05). (see *Figure 2—source data 2* for results of statistical analysis). While these results indicate general binding trends, they cannot provide $K_d$ estimates and do not have high enough resolution to distinguish between weaker binding interactions.

The online version of this article includes the following video, source data, and figure supplement(s) for figure 2:

**Source data 1.** Rolling parameter from all rolls for the indicated rollers on a DIDO1-PHD surface. Each row is a different roller and each column is an actuation.

**Source data 2.** Statistical analysis of results from METRIS measurements and histone peptide microarray results for DIDO1-PHD.

**Figure supplement 1.** Rolling parameter for all rolls on the DIDO-PHD surface.

**Figure 2—video 1.** Experimental video of H3.

https://elifesciences.org/articles/67525#fig2video1

**Figure 2—video 2.** Experimental video of H3K4me3-coated rollers on a DIDO1 PHD-coated substrate sped up 8X.

https://elifesciences.org/articles/67525#fig2video2

**Figure 2—animation 1.** Animation of 30 rolls for each roller for all DIDO1-PHD family of interactions: Orange, H3K9me3; Purple, H3K9me2; Cyan, H3K9me1; Green, H3; Grey; H3T3pK4me3.

https://elifesciences.org/articles/67525#fig2video3

---

(1200 µM ± 440). While these are empirically derived estimates for $K_d$, it is clear from the RP measurements that these interactions are statistically distinct, and they represent a missing piece of data that is fundamental to a quantitative understanding of epigenetic recognition.

The utility of the METRIS data is exemplified when evaluating the $ΔΔG^o$ (ΔΔG) values, a common way to report the energetic contributions of individual amino acids for a set of related PPIs. ΔΔG is calculated by taking the natural log of the ratio of two $K_d$ values ($K_{d1}$ and $K_{d2}$ in *equation 2*) in the Gibbs free energy equation, where R is the gas constant and T is the temperature in Kelvin (*Sidhu and Koide, 2007*).

$$ΔΔG^o(ΔΔG) = RT \ln \frac{K_{d1}}{K_{d2}} \qquad (2)$$

This analysis allows for calculating the energetic contributions of the individual PTMs for binding to the DIDO1-PHD domain. For example, K4me3 is worth $-4.2 \frac{kcals}{mol}$ while T3p is worth $+3.4 \frac{kcals}{mol}$ (*Figure 2D*). To our knowledge, this is the first energetic analysis of the DIDO1 phospho/methyl switch. These values have more context when viewed with the crystal structure of DIDO1-PHD (*Figure 2E*; *Gatchalian et al., 2013*). The hydrophobic trimethyl-lysine binding site accounts for a significant amount of the total binding to the peptide, however, there are clearly other residues on H3 that interact with DIDO1-PHD, such as the N-terminus, R2, and T3, and therefore, it is not surprising that unmodified H3 can still bind and account for roughly $-4 \frac{kcal}{mol}$ when using 1 M $K_d$ as the null reference. The deleterious effect of T3p is also resolved, since residue E308 of the PHD domain would clash and repel a T3p modified histone tail. Furthermore, this analysis also provides new insights into discrimination of methylation states by the DIDO1-PHD. For example, the greatest change in ΔΔG occurs between H3 from H3K4me1 ($-1.9 \frac{kcal}{mol}$), then H3K4me1 versus H3K4me2 ($-1.6 \frac{kcal}{mol}$), and H3K4me2 from H3K4me3 is the weakest ($-0.7 \frac{kcal}{mol}$). Thus, despite the DIDO1-PHD having the highest affinity for H3K4me3, it has the greatest discrimination between non-methylated H3K4 versus H3K4me1. The structure agrees with this observation, where two of the methyl binding sites are the most buried and the third is the most exposed one.

One of the significant advantages of the METRIS assay is that only 10 µl of 2 µM (20 pmol) is required to load the substrate and less is needed for the rollers, which is significantly less than any conventional method to measure PPI affinities. We compared METRIS to histone peptide microarrays, which is another methodology that can produce binding data with a minimal amount of protein (e.g. 500 µl of 0.5 µM [250 pmol] protein). While microarrays offer high-throughput screening, they lack the sensitivity to determine weak binding and small affinity differences. For DIDO1-PHD, we could observe a statistically significant difference between H3K4me3 and the other methylation

states, but there were no other statistically significant differences (*Figure 2F*, *Figure 2—figure supplement 1B*, and *Figure 2—source data 2*). Given this result, METRIS is significantly more sensitive and quantitative than other common methods to measure protein affinities that use comparable amounts of reagents at low concentrations (i.e. ELISA and microarrays).

## Determining ORC1-BAH methyl preferences using METRIS analysis

We further validated the METRIS assay using another methyllysine reader, the BAH domain of ORC1. ORC1 functions in licensing origins of replication by discriminating H4K20me2 from H4K20me1, a PTM on active chromatin, and H4K20me3 a repressive PTM (*Bicknell et al., 2011a*; *Bicknell et al., 2011b*; *Kuo et al., 2012*). We selected ORC1 because the reported affinities are within an order of magnitude, with a twofold difference reported between H4K20me1 and H4K20me3. The $\Delta x$ and RP values we obtained matched the published binding preferences (*Kuo et al., 2012*) H4K20me2 ($0.263 \pm 0.011$) > H4K20me1 ($0.226 \pm 0.008$) >H4K20me3 ($0.215 \pm 0.005$)> H4 ($0.202 \pm 0.005$) (*Figure 3A*, *Figure 3B*, *Figure 3—figure supplement 1A*, *Figure 3—video 1*, *Figure 3—video 2*, and *Figure 3—animation 1*). Using the same fitting method, we observe a linear log-log dependence ($R^2 = 0.967$) and the METRIS calculated $K_d$ values were between four- and eightfold tighter than the published values, yet there was good agreement between the fold-change and accordingly the $\Delta\Delta G$s. (*Figure 3C* and *Figure 3D*). Thus, the METRIS assay is sensitive enough to measure changes that are $0.4 \frac{kcal}{mol}$.

Using the METRIS assay, we could also measure binding to the unmodified H4, which has previously not been detected. H4 unmodified binding was measured to be 44-fold weaker than H4K20me2 binding. With this value we could calculate that the $\Delta\Delta G$ for K20me2 is worth $-2.2 \frac{kcal}{mol}$. When comparing this to the DIDO1-PHD, we find that DIDO1-PHD has a stronger interaction with the PTM ($-4.2$ versus $-2.2 \frac{kcal}{mol}$) however the ORC1-BAH domain has a stronger interaction with the unmodified histone than the DIDO1-PHD ($-6$ versus $-4 \frac{kcal}{mol}$). Examining the structure of ORC1-BAH domain bound to H4K20me2 (*Kuo et al., 2012*) shows the methyllysine binding pocket is more charged than DIDO1-PHD, and likely, in part, contributes to the higher affinity to the unmodified peptide (*Figure 3D*). The METRIS analysis also furthers our understanding of ORC1-BAH discrimination amongst methyl states. We find the greatest differentiation between H4K20me2 and H4K20me3 ($1.6 \frac{kcal}{mol}$) consistent with the biological role of ORC1 and this methyl sensing occurs through residue E93 (*Figure 3E*).

We also performed histone peptide microarrays on ORC1-BAH for comparison against the METRIS assay. The only statistically significant difference is between H4K20me2 and the other peptides (*Figure 3F* and *Figure 3—figure supplement 1B*), although the trends do match the literature and METRIS values, including the signal for the unmodified peptide when compared to the other peptides on the array, which support our findings with METRIS. However, due to the large standard deviation observed on the microarray, the assay would need to be repeated multiple times to achieve statistical significance. This highlights another advantage of METRIS assay, since it is a single particle method and the RP measurements are taken 36 times for each particle, this method has high statistical power.

## Investigating noncovalent interactions between Ubiquitin-like domains and Ube2D1 utilizing METRIS

We next used METRIS to investigate interactions with the protein post-translational modification ubiquitin. Ubiquitin has an expansive cellular regulatory role that is controlled by weaker interactions with effectors (*Oh et al., 2018*; *Cohen et al., 2017*) including non-covalent interactions with E2s and E3 ligases (*Brzovic et al., 2006*; *Zhang et al., 2016*). Ubiquitin binding is wide-spread, and there are hundreds of UBLs in the human genome for these readers to discriminate amongst (*Harrison et al., 2016b*). For example, the E2 Ube2D1 binds to ubiquitin noncovalently with an affinity of $206 \pm 6$ and we have shown that a ubiquitin-like domain (UBL) on the E3 UHRF1 can bind with higher affinity ($15 \pm 1$ μM with NMR or 29.0μM $\pm1$ with ITC) (*DaRosa et al., 2018*). To probe this interaction with METRIS, both ubiquitin and the UHRF1-UBL domain were labeled using biotin-PEG-maleimide at an N-terminal cysteine installed for labeling, and Ube2D1 was labeled at native cysteines. The $\Delta x$ and RP data match the affinity trend UHRF1-UBL ($0.131 \pm 0.005$) > ubiquitin ($0.108 \pm 0.004$) (*Figure 4A*, *Figure 4B*, *Figure 4—figure supplement 1A*, and *Figure 4—animation 1*) and

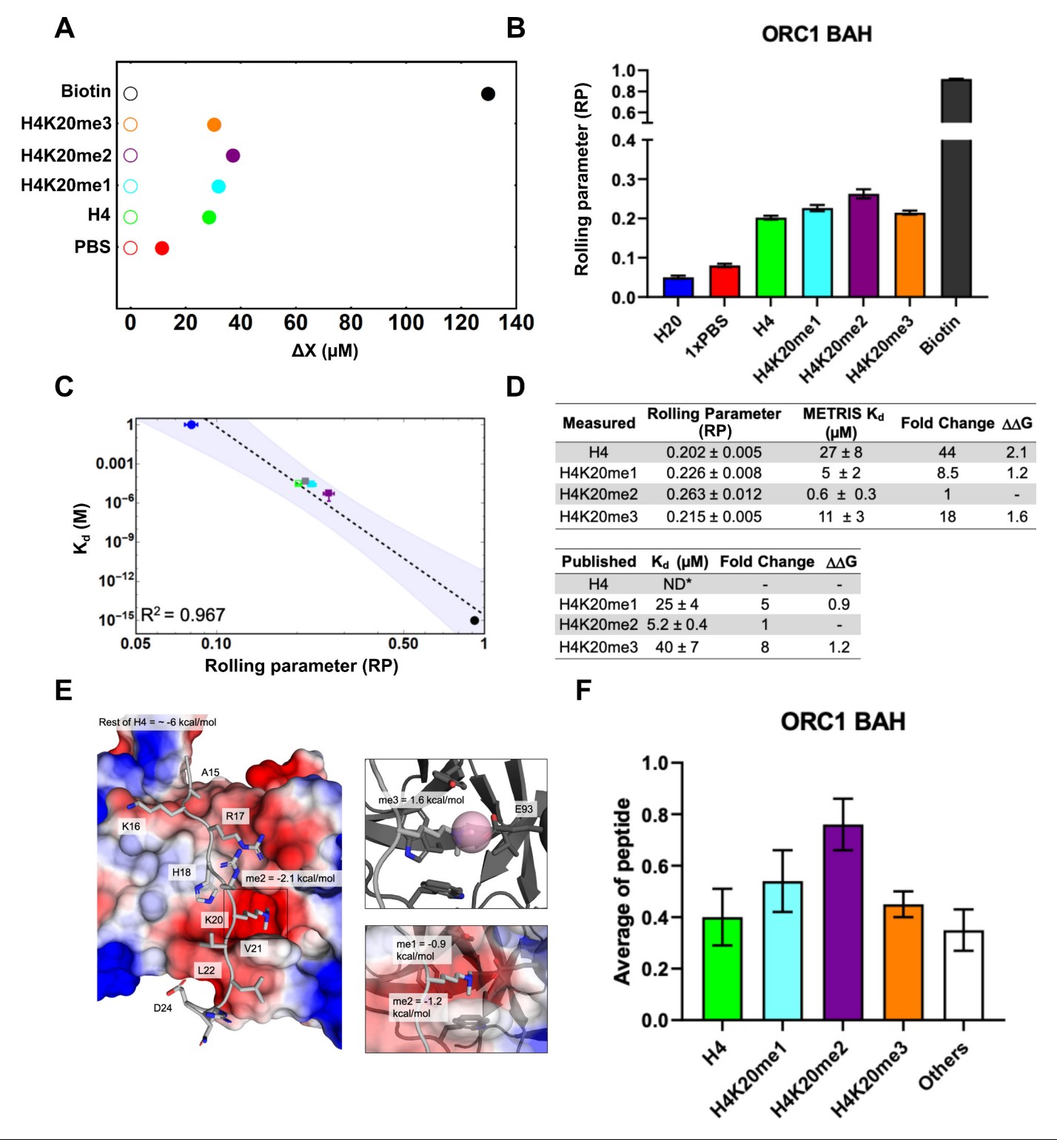

**Figure 3.** ORC1-BAH domain interactions characterized using METRIS. (**A**) Plot showing the average translational displacement per actuation for the rollers immobilized with the H4K20 methylated peptide on a ORC1-BAH domain surface. Streptavidin-Biotin and Streptavidin-avidin (PBS) are included for references. See *Figure 3—source data 2* for results of statistical analysis; all comparisons are significant (p<0.0001). (**B**) Plot showing the calculated RP for the indicated interactions (**C**) Log-Log plot of the rolling parameters, RP, from panel A. Extrapolated point markers are unfilled and the 95% $K_d$ confident interval for the fitting is depicted. (**D**) Table of rolling parameters and associated $K_d$ estimates for the ORC1-BAH. Fold change is calculated as the ratio between the $K_d$ for the indicated peptide and the $K_d$ for H4K20me2. These ratios are used to calculate $\Delta\Delta G$ at T=298K. The published

*Figure 3 continued on next page*

*Figure 3 continued*

values are from *Kuo et al., 2012* using ITC; *ND = Not determined. (E) Image of the ORC1-BAH crystal structure with H4K20me2 peptide, with the BAH surface electrostatic potentials shown (red = negative, blue = positive) as well as the $\Delta\Delta G$ for K20me2 and the estimate for the rest of the peptide. The PTM reader site is shown with greater detail to the right. Here the $\Delta\Delta G$ is calculated between the sequential methyl states. (F) Results of the ORC1-BAH histone peptide microarray assay against the indicated peptides from panel A (see *Figure 3—figure supplement 1B* for complete peptide plot). Only H4K20me2 is statistically significantly different (p<0.05) from the other H4 peptides (*Figure 3—source data 2*). Again, we see that microarrays can indicate general binding trends but they cannot provide $K_d$ estimates and do not have high enough resolution to distinguish between weaker binding interactions.

The online version of this article includes the following video, source data, and figure supplement(s) for figure 3:

**Source data 1.** Rolling parameter from all rolls for the indicated rollers on a ORC1-BAH surface. Each row is a different roller and each column is an actuation.

**Source data 2.** Statistical analysis of results from METRIS measurements and histone peptide microarray results for ORC1-BAH.

**Figure supplement 1.** Rolling parameter for all rolls on the ORC1-BAH surface.

**Figure 3—video 1.** Experimental video of H4-coated rollers on a ORC1-BAH-coated substrate sped up 8X.

https://elifesciences.org/articles/67525#fig3video1

**Figure 3—video 2.** Experimental video of H4K20me2-coated rollers on a ORC1-BAH-coated substrate sped up 8X.

https://elifesciences.org/articles/67525#fig3video2

**Figure 3—animation 1.** Animation of 30 rolls for each roller for all ORC1-BAH family of interactions: Orange, H4K20me3; Purple, H4K20me2; Cyan, H4K20me1; Green, H4.

https://elifesciences.org/articles/67525#fig3video3

fitting METRIS-$K_d$s produced values that were 5-fold weaker than the published values, but were in exact agreement with the 13-fold difference (1.4 $\frac{kcal}{mol}$ G) reported in the literature (*Figure 4C*). Therefore we have demonstrated that METRIS can measure and distinguish interactions in the $10^{-4}$ M range without utilizing highly concentrated protein solutions, providing a simple method to measure weak interactions.

## Direct measurement of an interdomain interaction between UBL and SRA domains of UHRF1 using METRIS

For epigenetic readers/writers, there is an abundance of examples where interdomains interactions within a single polypeptide chain control allostery (*Worden et al., 2019*; *Ruthenburg et al., 2007*). For example, the role of UHRF1 in controlling DNA methylation requires interactions between its domains (*Harrison et al., 2016a*; *Gao et al., 2018*; *Gelato et al., 2014*), and specifically, our previous study provided evidence for an interaction between the UHRF1-UBL and the UHRF1-SRA domain, which is required for ubiquitylation of histone H3 (*DaRosa et al., 2018*). Studying interdomain interactions can be difficult, given the weak and transient nature of these interactions. We therefore thought METRIS is well-suited to measure this type of interaction. Accordingly, we tested the SRA and UBL interaction with METRIS by attaching biotinylated SRA to the substrate. For the SRA-UBL interaction, we measured an RP of 0.119 ± 0.004 for the particles, significantly higher than the 0.081 for an unmodified surface and the 0.085 we obtain with ubiquitin on the roller (*Figure 4A* and *Figure 4B*). This represents the first direct measurement of the interaction between the SRA and UBL domains of UHRF1. We also tested a mutation to the UBL (W2V) that previous biochemical assays suggested is critical for the interaction (*DaRosa et al., 2018*; *Foster et al., 2018*), and W2V had a significantly reduced RP to 0.098 ± 0.002 (*Figure 4A*, *Figure 4B*, and *Figure 4—figure supplement 1A*). Fitting METRIS-$K_d$ shows the $\Delta\Delta G$ of the W2V variant is worth 1.5 $\frac{kcal}{mol}$ (*Figure 4D*) due to replacing the aromatic sidechain with the short aliphatic side chain (*Figure 4E*). This highlights another strength of METRIS; it is rare to assign $\Delta\Delta G$ values to mutations at binding hotspots because the mutated variant binds weakly (*Pál et al., 2006*). Therefore, we expect that METRIS will greatly enhance our understanding of PPIs.

## Global fit of METRIS analysis

We sought to generate a global fit for all of the measurements from the three independent data sets. Overall, the log-log fit of the data remained linear ($R^2 = 0.89$) (*Figure 5A*), and even using this global fit, we observe agreement between fold changes and $\Delta\Delta G$ within a given set of PPIs (*Figure 5B*). However, the METRIS-$K_d$ values were less accurate than with the individual fitting and

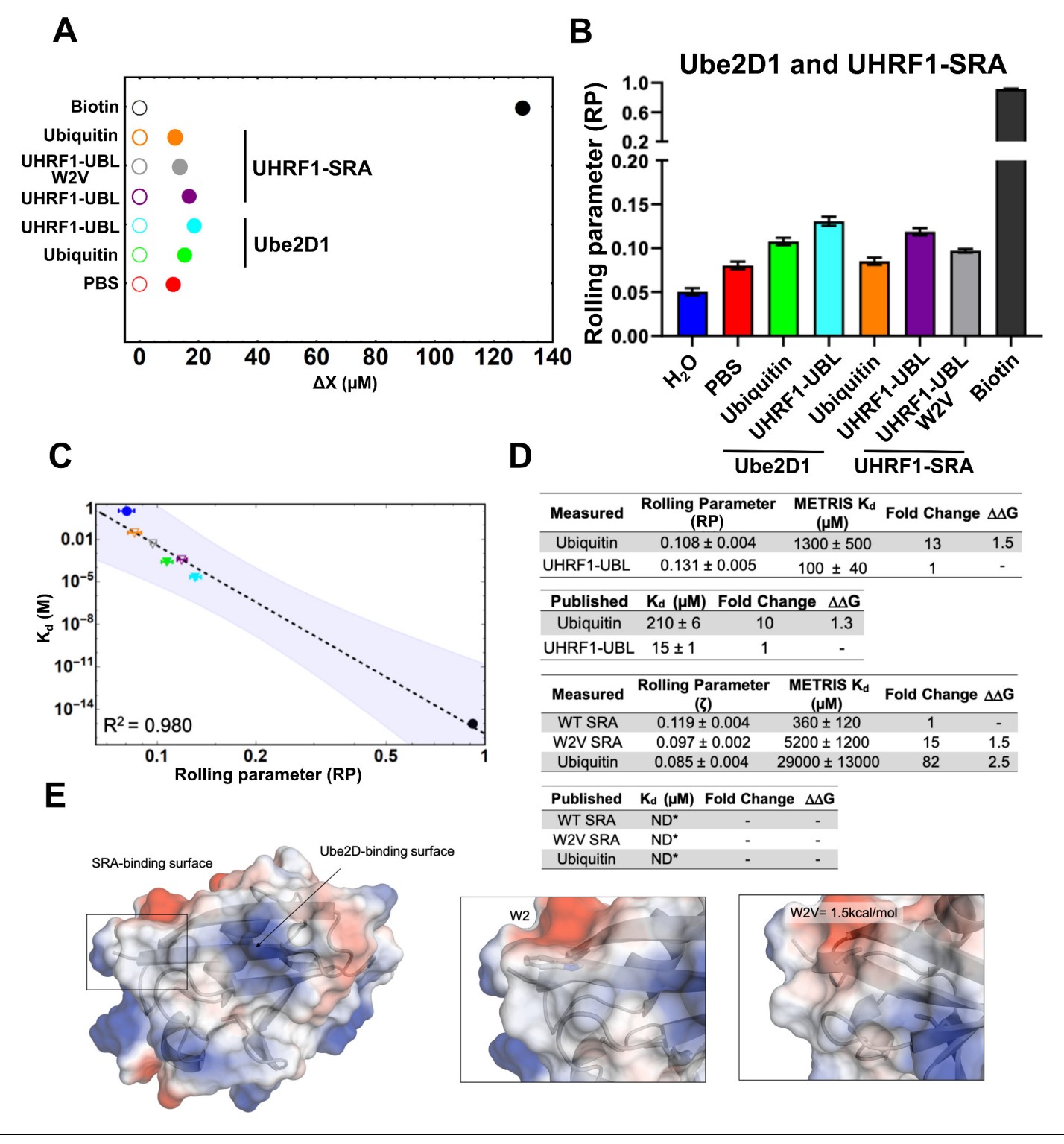

**Figure 4.** Measuring the interaction of UBL domains with Ube2D1 or the UHRF1-SRA domain using METRIS. (**A**) Plot showing the average translational displacement per actuation for the rollers coated with the indicated proteins on a surface containing UbeD21 (E2) or the UHRF1-SRA domain. Streptavidin-Biotin and Streptavidin-avidin (PBS) are included for references. All comparisons are statistically significant (See *Figure 4—source data 2* for results of statistical analysis.) (**B**) Plot showing the calculated RP for the indicated rollers (**C**) Log-Log plot of the rolling parameters, RP, from panel A. Extrapolated point markers are unfilled and the 95% confident interval for the fitting is depicted. (**D**) Table of all rolling parameters and associated METRIS-$K_d$ estimates. Fold change is calculated as the ratio between the indicated protein and the UHRF1-UBL domain and the $\Delta\Delta G$ is calculated using these ratios at T=298K. $K_d$ values for ubiquitin are taken from *Buetow et al., 2015* and UHRF1-UBL value taken from *DaRosa et al., 2018*. (**E**)

*Figure 4 continued on next page*

*Figure 4 continued*

Image of the UHRF1-UBL binding surface for the UHRF1-SRA and Ube2D1 shown with electrostatic surface potentials (red = negative, blue = positive) with insets highlighting the change of the UBL surface with the W2V mutation and the associated $\Delta\Delta G$.

The online version of this article includes the following source data and figure supplement(s) for figure 4:

**Source data 1.** Rolling parameter from all rolls for the indicated rollers on either a Ube2D1 surface or a UHRF1-SRA surface.
**Source data 2.** Statistical analysis of results from METRIS measurements for ubiquitin and UHRF1-UBL binding to Ube2D1 or UHRF1-SRA domain.
**Figure supplement 1.** Rolling parameter for all rolls on the Ube2D1 or SRA surface.
**Figure 4—animation 1.** Animation of 30 rolls for each roller for all Ube2d1 and UHRF1 UBL family of interactions: Cyan, UHRF1-UBL; Green, ubiquitin; both on a Ube2D1 surface.

https://elifesciences.org/articles/67525#fig4video1

we could not discriminate between similar strength binders in different sets of PPIs (e.g. between DIDO1 and ORC1). These results indicating that we cannot directly compare RP values obtained for different types of PPIs and that there is likely some structural difference in each system that is not yet accounted for. However, given that each set of values had similar systematic deviations from the experimentally determined values, which is why the $\Delta\Delta G$ remained accurate, we realized we could

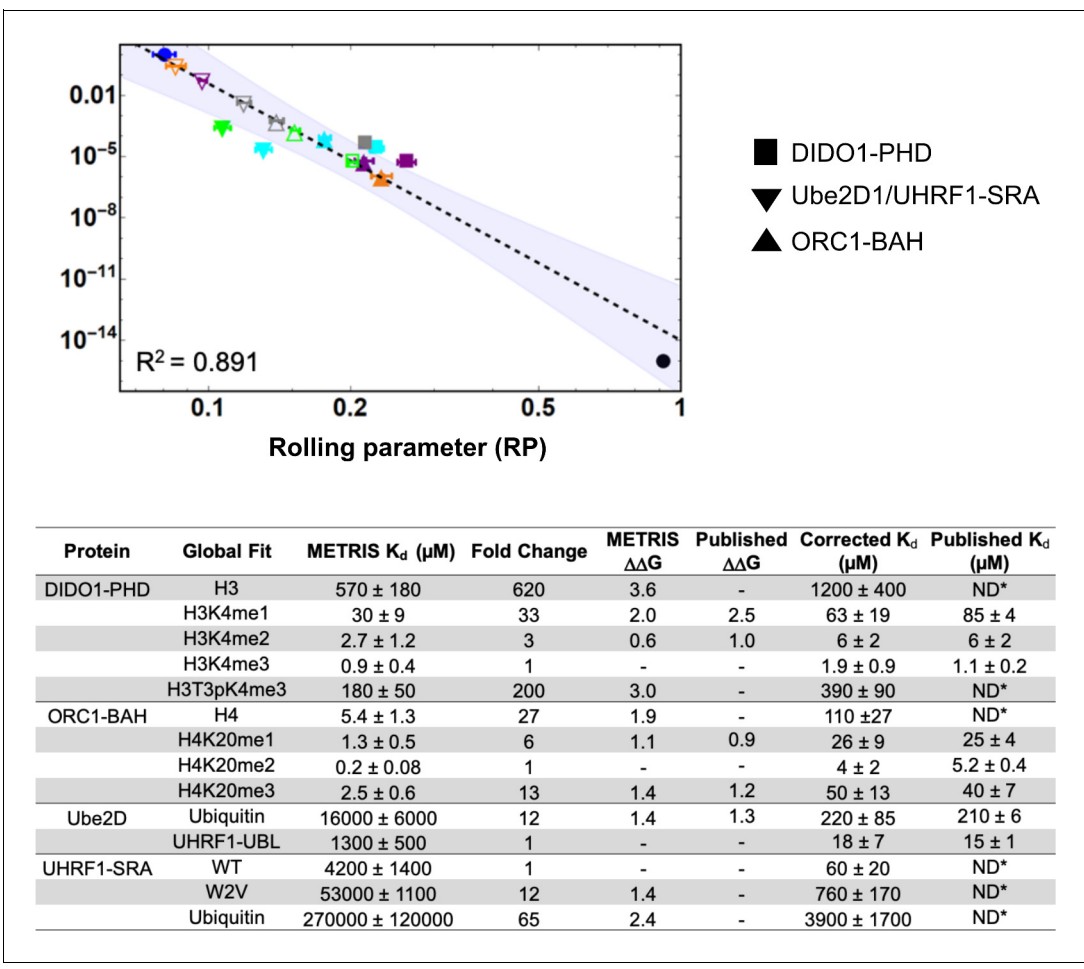

| Protein | Global Fit | METRIS $K_d$ (µM) | Fold Change | METRIS $\Delta\Delta G$ | Published $\Delta\Delta G$ | Corrected $K_d$ (µM) | Published $K_d$ (µM) |
|---|---|---|---|---|---|---|---|
| DIDO1-PHD | H3 | 570 ± 180 | 620 | 3.6 | - | 1200 ± 400 | ND* |
| | H3K4me1 | 30 ± 9 | 33 | 2.0 | 2.5 | 63 ± 19 | 85 ± 4 |
| | H3K4me2 | 2.7 ± 1.2 | 3 | 0.6 | 1.0 | 6 ± 2 | 6 ± 2 |
| | H3K4me3 | 0.9 ± 0.4 | 1 | - | - | 1.9 ± 0.9 | 1.1 ± 0.2 |
| | H3T3pK4me3 | 180 ± 50 | 200 | 3.0 | - | 390 ± 90 | ND* |
| ORC1-BAH | H4 | 5.4 ± 1.3 | 27 | 1.9 | - | 110 ± 27 | ND* |
| | H4K20me1 | 1.3 ± 0.5 | 6 | 1.1 | 0.9 | 26 ± 9 | 25 ± 4 |
| | H4K20me2 | 0.2 ± 0.08 | 1 | - | - | 4 ± 2 | 5.2 ± 0.4 |
| | H4K20me3 | 2.5 ± 0.6 | 13 | 1.4 | 1.2 | 50 ± 13 | 40 ± 7 |
| Ube2D | Ubiquitin | 16000 ± 6000 | 12 | 1.4 | 1.3 | 220 ± 85 | 210 ± 6 |
| | UHRF1-UBL | 1300 ± 500 | 1 | - | - | 18 ± 7 | 15 ± 1 |
| UHRF1-SRA | WT | 4200 ± 1400 | 1 | - | - | 60 ± 20 | ND* |
| | W2V | 53000 ± 1100 | 12 | 1.4 | - | 760 ± 170 | ND* |
| | Ubiquitin | 270000 ± 120000 | 65 | 2.4 | - | 3900 ± 1700 | ND* |

**Figure 5.** Global fit of binding partners for all METRIS experiments performed. (**A**) Global Log-Log plot showing linearity between rolling parameter and dissociation constant for all interactions measured. (**B**) Table of binding constants of tested interaction partners when determined from the global fit. Fold change and $\Delta\Delta G$ are calculated in the same way as the previous example and the $\Delta\Delta G$ are similar to the previous reported value. Scaling factors are calculated by averaging the fold difference between METRIS-$K_d$ and the published $K_d$ for all interactions of the same type. Then the METRIS-$K_d$ is multiplied by the scaling factor yielding the corrected $K_d$s, which match the published values.

apply a simple scaling factor to the METRIS-$K_d$ values to obtain measurements that matched the experimentally determined $K_d$. To determine the scaling factors for each interaction, we divided the published value against the METRIS-$K_d$, and averaged them, and then multiplied the METRIS-$K_d$ by the scaling factor and could reproduce the literature values (*Figure 5B*). Indeed, this scaling can be applied to each of the individual fits to reproduce the literature values. It also provides a simple way to scale METRIS-$K_d$ values to any experimentally determined $K_d$ values.

## Discussion

METRIS, which measures the effective mechanical friction induced by PPIs, is fundamentally different than current methodologies. Here, we have shown that METRIS can be advantageous to other approaches when measuring weak interactions, since both binding partners are immobilized. Thus, METRIS uses a very low concentration of proteins while maintaining high precision. These characteristics allow for the characterization of a vast array of PPIs, many of which were previously very laborious to measure. In this study, we demonstrate how METRIS can contribute to the study of epigenetics, by allowing us to assign $\Delta\Delta G$ for PTMs individually and in combination, including a phospho/methyl switch in DIDO1. These values are significant because they provide a quantitative measure for the interplay between concurrent PTMs, a central premise of the epigenetic code (*Rothbart and Strahl, 2014*). Furthermore, this study shows that even applying METRIS to characterized interactions can provide new insights into PPIs.

Another area where better characterization of weak interactions will contribute significantly to understanding is in studying interdomain interactions. These types of interactions can be difficult to quantify without very resource-intensive processes, and limitations with the proteins themselves (yield or solubility) may make these interactions unmeasurable. Currently, pulldown assays, chemical crosslinking, and proximity ligation are qualitative, rarely produce quantitative data, and require mass spectrometry (*Richards et al., 2021*). Here, we have measured a direct interaction between the UHRF1-UBL and SRA domains that we estimate to have a $K_d$ 60µM (*Figure 5B*), however, the biological context for this interaction is between two tethered domains, so an absolute value is only partially relevant. More generally, we show that METRIS can be used to measure $\Delta\Delta G$ for hotspot mutations, which to our understanding, could previously only be measured indirectly using high-throughput selection strategies (*Pál et al., 2006*). Thus, METRIS will provide additional new data to the field of protein biochemistry and could aid in the parametrization of computational binding score functions.

METRIS can be conceptualized as a measure of the ability of converting rotational motion into translation motion. To translate, there must be sufficient magnetic torque applied to the rollers such that the PPI or other interactions between the roller and the substrate can be broken as they translate across the surface. Previous studies using force spectroscopy methods have measured the force of the biotin-streptavidin interaction to be 160pN (*Florin et al., 1994*). However, while there are some similarities with these techniques and METRIS (e.g. single particles in both cases *Neuman and Nagy, 2008*), there are distinct differences, particularly with the use of magnetic torque to impart this rolling motion in this nonequilibrium active system. Still, the effective friction is due to these intermolecular forces, but currently we do not have a model for how to relate the force directly to the rolling.

An essential advantage of METRIS is resolution, precision, and sensitivity, which allows for the differentiation of $\Delta\Delta G$ values as small as 0.4 $\frac{kcal}{mol}$. Several factors likely contribute to this robustness: (1) the rolling parameter is not inherently dependent on the protein concentration, so long as the rollers and surface are saturated. (2) The measurements have high statistical power ($\approx 10$ particles each with 36 RP measurements) and very low percentage error, given the high accuracy of the measurement and low uncertainty of many of the variables in the rolling parameter calculation. (3) Multiple interactions between the bead and the surface amplify the friction, which may be necessary to measure weak interactions, and likely limits the impacts of inactive proteins on the roller and substrate. However, METRIS does have limitations, such as the reliance on literature values for extrapolating and scaling the METRIS-$K_d$ and the need to biotinylate the binding proteins. We envision with future development, we will derive a better mathematical model that describes the relationship between protein affinity and rolling parameter as many factors will contribute to the friction, such as the

number of interactions per bead or the size of the protein interaction. Despite these limitations, we expect that METRIS will be of great use to researchers studying PPIs.

# Materials and methods

**Key resources table**

| Reagent type (species) or resource | Designation | Source or reference | Identifiers | Additional information |
|---|---|---|---|---|
| Recombinant DNA reagent | BirA-GST-Orc1-BAH; Orc1-BAH | This Study | | Strahl Lab, pGEX vector, *Figure 2* |
| Recombinant DNA reagent | BirA-GST-DIDO1-PHD; DIDO1-PHD | This Study | | Strahl Lab, pGEX vector, *Figure 3* |
| Recombinant DNA reagent | N-cys Ubiquitin; ubiquitin | Kamadurai, Hari B et al. 'Insights into ubiquitin transfer cascades from a structure of a UbcH5B approximately ubiquitin-HECT (NEDD4L) complex.' Molecular cell vol. 36,6 (2009): 1095–102. | | pGEX vector, |
| Recombinant DNA reagent | Ube2D1 | DaRosa, Paul A et al. 'A Bifunctional Role for the UHRF1 UBL Domain in the Control of Hemi-methylated DNA-Dependent Histone Ubiquitylation.' Molecular cell vol. 72,4 (2018): 753–765.e6. | | Pet15 vector, |
| Recombinant DNA reagent | UHRF1-SRA | This study | | Harrison Lab, MBP-pQ80L, *Figure 4* |
| Recombinant DNA reagent | N-cys-UHRF1-UBL W2V | This study | | Harrison Lab, MBP-pQ80L, *Figure 4* |
| Recombinant DNA reagent | N-cys-UHRF1-UBL | This study | | Harrison Lab, MBP-pQ80L, *Figure 4* |
| Software | Able Particle Tracker | Mu Labs | Full Version | http://apt.mulabs.com/ |
| Software | Mathematica | Steimel Labs | | |
| Software | OGG Video Converter | Ogg-converter.net | Version 6 | |
| Software | Pymol | Schrödinger | V2.4 | |
| Software | Prism | GraphPad | | |
| Other | Streptavidin Coated Ferromagnetic Beads | Spherotech | SVFM-100–4 | |
| Other | Avidin Coated Slides | ArrayIt | SMV | |
| Other | Histone peptide array | Petell, Christopher J et al. 'Improved methods for the detection of histone interactions with peptide microarrays.' Scientific reports vol. 9,1 6265. 18 Apr. 2019 | | |

*Continued on next page*

*Continued*

| Reagent type (species) or resource | Designation | Source or reference | Identifiers | Additional information |
|---|---|---|---|---|
| Antibody | Anti-GST (rabbit polyclonal) | Epicypher | 13–0022 | (1:1000) |
| Antibody | Anti-Rabbit AlexaFluor-647 (goat polyclonal) | Invitrogen | A21244 | (1:10,000) |

## Magnetic probe and substrate functionalization

The streptavidin-coated ferromagnetic particles, provided by Spherotech with a nominal diameter of 10 µm, are composed of a core of polystyrene and $CrO_2$. 10 µL of the stock solution, 1.0% w/v, was extracted inserted into a micro-centrifuge tube. Biotinylated peptides were then inserted into the tube with the streptavidin-coated ferromagnetic particles. The amount of peptides was such to coach each bead $50\times$ the theoretical limit, 1 mg of beads binds 0.18 nmole of biotin, to ensure all binding sites on the beads were covered. The bead and peptide solution was left to react at room temperature for at least 2 hr. The density of biotin binding sites per roller is on the order of $6 \times 10^8$ biotin molecules per roller which corresponds to a density of biotin binding sites on the order of $5 \times 10^{10}$ binding sites per mm$^2$. Assuming the footprint of the sphere on the substrate to be approximately 1% of the projected area that would result in approximately $10^5$ potential binding sites per bead, although the actual number of binders is likely much lower due to steric hindrance and other effects. The substrates are avidin-coated glass slides, provided by Arrayit, with a ligand density of $1.1 \times 10^{10}$ ligands per mm$^2$. Microfluidic channels were created on this substrate using two pieces of double-sided tape, provided by 3M. The pieces of double-sided tape were cut to a width of several mms and a length of at least 25 mm. The pieces of tape were placed parallel to each other and at a distance of approximately 3–4 mm apart. Then a glass coverslip was placed on top of the tape to create channels approximately $22 \times 5$ mm. A solution of biotinylated proteins was then inserted into the channel. The amount of proteins inserted was enough to coat the channel surface $50\times$ the theoretical limit to ensure that all of the sites on the substrate were coated. The substrate and solution was left in a container for 2 hr to allow the proteins time to bind to the substrate. After 2 hr, the solution was washed from the channel to remove any excess protein that was not attached to the substrate. Then the solution of peptide coated ferromagnetic beads was diluted approximately $2000\times$ to reduce the probability of two ferromagnetic beads forming a magnetic dimer which cannot be analyzed in the rolling parameter analysis and dimer particles are excluded. The channel was sealed with epoxy and magnetized by an external permanent neodymium magnet. The substrate was placed in the slide holder at the center of the Helmholtz Coil Inspired Experimental Apparatus.

## Helmholtz coil inspired experimental apparatus

The Helmholtz Coil Inspired Apparatus consists of three pairs of coils were secured in an apparatus, made of aluminum T-slots, and attached to an optical breadboard. The coils have an inner diameter of 7 cm and an outer diameter of 13 cm, as seen in *Figure 1—figure supplement 1*. Two sinusoidal signals, phase shifted by 90 degrees, were generated in Matlab. Those signals were sent to a National Instruments USB X Series DAQ and then passed through a 300W amplifier (150W/channel) before being sent through each pair of coils. The magnetic field is large enough (approximately 10mT) to ensure alignment of the rotational frequency of the particles with the frequency, $\omega$, of the magnetic field. The signal from the amplifier was routed to a Rigol Oscilloscope to measure the frequency and the voltage. The sample holder is made from 6061 aluminum and attached to a OMAX binocular microscope which functions as our 3D optical stage. Data acquisition was accomplished via a CMOS camera mounted on a C-mount DIN objective tube assembly. The camera was connected to a computer for visualization, video capture, and subsequent analysis. As mentioned, the magnetic field strength (B) Is approximately 10mT and the ferromagnetic particles exhibit a magnetic moment (m) on the order of $10^{-11}$ Am$^2$. This combination of magnetic field strength and magnetic moment allows for sufficient magnetic torque, $\tau$ to be applied to break the strongest non-covalent interaction, biotin and streptavidin.

$$\tau = m \times B \tag{3}$$

For these experiments, once the sample was in the apparatus the actuation protocol was as follows: a five second actuation period, $\tau$ in the RP equation, where the field rotates clockwise at a rotational frequency, $\omega$, of 1 Hz. The field is then shut off for 5 s to allow the particle to settle and bonds to equilibrate, and then the field rotated counter-clockwise for the same actuation period and at the same rotational frequency, and then the field is shut off again for 5 s. After this the process repeats 18 times, after which the video is post-processed and the particles are tracked.

## Particle tracking

To analyze particle motion, we converted the video captured from the CMOS camera and converted it into an .avi file using ArcSoft Media Converter eight and then converted the .avi into a sequence of .jpeg images. To analyze the motion of the active particles we used Able Particle Tracker. The data was then imported into Mathematica for subsequent analysis. Particles that stick together, due to attraction between the ferromagnetic beads, are omitted from the analysis. The custom Mathmatica scripts measures the diameter of each roller and the distance that the particle travels per actuation period and calculates the rolling parameter according. The recorded images have a resolution of approximately 1 µm per pixel, and our image analysis software has subpixel resolution so we can measure differences in displacement on the order of 0.1 µm.

## Protein purification and biotinylation

GST-[DIDO1-PHD/ORC1-BAH]-avi recombinant proteins we cloned into the pGEX-4T1 vector (GE, 27458001) to generate GST-[DIDO1-PHD/ORC1-BAH]-avi recombinant proteins. Recombinant proteins were purified as described in previous work . Briefly, the recombinant proteins were induced to express in SoluBL21 cells (Fisher, C700200) after reaching an OD600 of 0.4 with 0.2 mM IPTG and by shifting to 16°C for overnight growth. After induction, the cells were pelleted and resuspended in Lysis Buffer (50 mM HEPES, 150 mM NaCl, 1 mM DTT, 10% glycerol, pH 7.5) supplemented with protease inhibitors, then incubated in the presence of lysozyme (Sigma, L6876) and nuclease (ThermoFisher, PI88700) for 30 min. After this the cells were sonicated for six rounds consisting of 10 s continuous sonication at 50% intensity, 50% duty cycle followed by 60 s on ice. Lysates were centrifuged for 10 min at 10,000 rpm and the clarified lysates loaded onto a glutathione resin and purified by batch purification according to the manufacturer's protocol (ThermoFisher, PI16101). Purified proteins were then dialyzed against Lysis Buffer to remove GSH and quantified using a Bradford assay per the manufacturer instructions (BioRad, 5000006) prior to being stored at −80°C. Ube2D1 is a his-tagged protein that was purified according to previous publications through standard Ni-NTA purification. The UHRF1-UBL, W2V-mutant, and UHRF1-SRA domain were cloned into a modified version of His-MBP-pQE80L vector that we have previously described. For the UHRF1-UBL domain and W2V mutant an N-terminal cystine was added using PCR for chemical conjugation with maleimide. These proteins were grown to O.D. 0.6 and induced with 0.6 mM IPTG. MBP was cleaved using TEV purified in house and removed using anion exchange. The ubiquitin with an N-terminal cystine was purified using a pGEX-4T1 expression system described previously. The ubiquitin was removed from the resin by cleavage with TEV. Purified proteins with an avi-tag were biotinylated by using BirA following the BirA500 kit's protocol (Avidity, BirA500). Biotinylation was confirmed by performing a Coomassie gel shift assay according to Fairhead and Howarth, 2015. Cysteine Biotinylation was carried out using Poly(ethylene glycol) [N-(2-maleimidoethyl)carbamoyl]methyl ether 2-(biotinylamino)ethane (Sigma 757748) (Biotin-maleimide). Typically, small volumes were biotinylated such that very little biotin-malamide was needed (below a mg) so we added some powder and confirmed biotinaylation with SDS-page gel. For UHRF1-UBL variants and ubiquitin there is only a single engineered cysteine available for modification. For the Ube2D, UHRF1-SRA domain, and GST-PHD-DIDO1, we labeled native cysteines which resulted in heterogenous labeling. Excess biotin-maleimide was removed using size-exclusion or anion exchange for the UHRF1-UBL and ubiquitin, and dialysis for the SRA and GST-PHD-DIDO1. Proteins were typically aliquoted and frozen before MET-RIS analysis. Both labeling methods (N-terminal BirA tag versus biotin-maleimide) were evaluated for their ability to return RP values within error, which is shown in *Figure 2—figure supplement 1C*.

## Histone peptide microarrays

Histone peptide microarrays were performed and analyzed as described in *Petell et al., 2019*. In brief, 500 nM of the avi- and GST-tagged DIDO1-PHD or ORC1-BAH constructs in 1% milk 1x PBST (10 mM $Na_2HPO_4$, 1.8 mM $KH_2PO_4$, 2.7 mM KCl, 137 mM NaCl, pH 7.6, 0.1% Tween-20) were incubated overnight at 4°C with shaking. The following day, the arrays were washed by submerging in 1x PBS briefly, then submerged in 0.1% formaldehyde in 1x PBS for 15 s to cross-link, formaldehyde was then quenched by submerging in 1 M glycine in 1x PBS for 1 min, after which the arrays were submerged in 1x PBS and inverted five times to remove remaining glycine. Next, the arrays were washed three times with high-salt 1 X PBS (1x PBS with 497 mM NaCl rather than 137 mM NaCl) for 5 min each at 4°C with shaking. Then, the arrays were incubated with a 1:1000 dilution of anti-GST (EpiCypher, 13–0022) in 1% milk 1x PBST for two hours at 4°C with shaking. After incubation with anti-GST antibody the arrays were washed with 1x PBS, three times for five minutes at 4°C with shaking. Next, they were exposed to a 1:10,000 dilution of anti-Rabbit AlexaFluor-647 (Invitrogen, A21244) for 30 min at 4° with shaking. Lastly, the arrays were washed three times for 5 min with 1x PBS as in the previous wash step, then submerged in 0.1x PBS prior to imaging. The arrays were imaged using a Typhoon (GE) and quantification was carried out using ImageQuant TL software. Analysis of the data was done by first averaging the triplicate intensities for a given peptide on the array; the values for an arrays' dataset were then linearly scaled from 0 to 1 by applying a min-max formula such that the minimum value became 0 and the maximum 1. After, this all the scaled array values were combined to derive a single average and standard deviation for each peptide and the averages used for the graphs; see plots for what peptide modification states are shown. Results for the DIDO1-PHD and ORC1-BAH domains showing all peptides carrying the specified modifications, alone and in combination with other PTMs is shown in *Figure 2—figure supplement 1C* and *Figure 3—figure supplement 1B*.

## Data collection and statistical analysis

All experiments for METRIS measured at least 10 different rollers that were rolled 36 times and all array data consist of at least three replicates, and averages with standard deviation are shown in the tables for each figure. All statistical analyses were done by using the Student's T-Test (unpaired, two-tailed distribution) using Graphpad Prism. The results of this statistical analysis are reported in *Figure 2—source data 2*, *Figure 3—source data 2*, and *Figure 4—source data 2*.

## Acknowledgements

This work was supported by NIH grant GM126900 to BDS, an ACS Postdoctoral Fellowship (PF-19-027-01–DMC) to CJP University of the Pacific startup funds to JPS and JSH and an undergraduate research grant to KR.

## Additional information

### Competing interests

Joshua P Steimel: is a cofounder of Tribosense Technologies. The other authors declare that no competing interests exist.

### Funding

| Funder | Grant reference number | Author |
|---|---|---|
| University of the Pacific | | Joseph S Harrison<br>Joshua P Steimel |
| National Institute of General Medical Sciences | GM126900 | Brian D Strahl |
| American Cancer Society | PF-19-027- 459 01-DMC | Christopher J Petell |

The funders had no role in study design, data collection and interpretation, or the decision to submit the work for publication.

## Author contributions
Christopher J Petell, Conceptualization, Resources, Data curation, Formal analysis, Investigation, Visualization, Writing - original draft, Writing - review and editing; Kathryn Randene, Michael Pappas, Resources, Data curation, Formal analysis, Investigation; Diego Sandoval, Data curation, Formal analysis, Investigation; Brian D Strahl, Conceptualization, Resources, Formal analysis, Supervision, Funding acquisition, Methodology, Writing - original draft, Writing - review and editing; Joseph S Harrison, Conceptualization, Resources, Data curation, Formal analysis, Supervision, Investigation, Visualization, Methodology, Writing - original draft, Project administration, Writing - review and editing; Joshua P Steimel, Conceptualization, Resources, Data curation, Software, Formal analysis, Supervision, Funding acquisition, Validation, Investigation, Visualization, Methodology, Writing - original draft, Project administration, Writing - review and editing

## Author ORCIDs
Brian D Strahl (iD) http://orcid.org/0000-0002-4947-6259
Joseph S Harrison (iD) https://orcid.org/0000-0002-2118-6524
Joshua P Steimel (iD) https://orcid.org/0000-0003-2437-8545

## Decision letter and Author response
Decision letter https://doi.org/10.7554/eLife.67525.sa1
Author response https://doi.org/10.7554/eLife.67525.sa2

# Additional files

## Supplementary files
• Transparent reporting form

## Data availability
We have included all of the calculated rolling parameters for each roll as source data and movies and animations of the experimental results are included in the manuscript.

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
