## [Decision Letter]

**Acceptance summary:**

This article introduces a new experimental method that enables the direct
measurements of weak interactions between proteins. It is based on densely attaching
one type of protein (antigen) to a ferromagnetic microsphere, which can then bind to
another protein that has been attached to a flat surface, and applying an external,
rotating magnetic field to force the particle to roll across the surface, where its
motion is slowed by binding and unbinding of antigens. The sensitivity of the method
suggests that it may prove useful in the study of weak protein-protein
interactions.

**Decision letter after peer review:**

Thank you for sending your article entitled "Mechanically Transduced
Immunosorbent Assay To Measure Protein-Protein Interactions" for peer review at
*eLife*. Your article is being evaluated by 2 peer reviewers, and
the evaluation is being overseen by a Reviewing Editor and Cynthia Wolberger as the
Senior Editor. The following individual involved in reviewer of your submission has
agreed to reveal their identity: Erika Eiser (Reviewer #3).

As you can see from the reviews below, the reviewers agree that the method you have
described has interesting potential, particularly in the small amount of material
needed for the assay. However, there are concerns about the fact that the method
itself has been published already some time ago, lessening the novelty of the
publication. Focusing on the applications of the method would in principle be
acceptable, but the concerns about the dynamic range of the method, and the
important need for validation need to be addressed (perhaps requiring new
experiments to do so) before we can reach a decision on your paper.

*Reviewer #1:*

The authors are seeking to develop a method for measuring the binding affinities
between biomolecules using low amounts of material. The method described in this
manuscript uses a magnetic field to cause a ferromagnetic particle to roll along a
surface. When the particle is modified with one biomolecule and the surface is
modified with another, there is an increase in friction if the two molecules
interact. The degree of interaction is proportional to the amount of friction.

Strengths

– The method requires a minimal amount of material.

– The rolling parameter can be measured for many ferromagnetic particles
simultaneously, leading to robust statistical analyses.

Weaknesses

– The method itself has already been described in a paper published by one of the
corresponding authors in 2014.

– Insufficient detail is provided on the method and output. Raw data showing that the
rolling particle is displaced in the presence of a binding partner on the surface is
missing.

– The Kd cannot be directly obtained from the rolling parameter.

– Unlike methods such as surface plasmon resonance and biolayer interferometry, which
use a comparably low amount of material, this method requires both binding partners
to be affinity-tagged. The tags and the immobilization could obscure native
interactions.

– Biomolecular interactions have a wide range of affinities that span over 15 orders
of magnitude (1 M to 10-15 M). Yet, the dynamic range of the rolling parameter is
only one order of magnitude (0.081 for Kd of 1 M and 0.918 for Kd of 10-15M). As a
result, the rolling parameter values cluster together when correlated with Kds on a
logarithmic scale. Subtle differences in binding affinity can therefore be lost.

– For any given system, the Kd must be measured using an orthogonal technique in
order to establish binding affinities that can be correlated with the measured
rolling parameter. More specifically, the rolling parameter values are not
generalizable from system to system.

– The schemes presented in Figure 1 are difficult to interpret. Perhaps it would be
better show a video of the particles rolling along a surface with no binding partner
and another video when there is a binding partner. In the absence of these raw data
for each system, the binding measurements are not compelling.

– The authors spend a great deal of time comparing their method to NMR, why? Other
measurements such as SPR and BLI would be more appropriate comparisons.

– In general, I found the way in which the manuscript was written to be very
misleading. After reading the Intro, I was expecting the rest of the paper would
focus on the development of the method. Then, in lines 184-185 on page 6, it becomes
clear that the method was already published 7 years ago by one of the corresponding
authors. Thus, the paper is less about the method and more about the
application.

– Please remove sentences like "It is difficult to overstate how transformative
this will be for the study of PPIs." The method is not a direct measurement of
binding affinity. The dynamic range of the measurement does not correlate with the
range of biomolecular binding affinities. Multiple affinity tags must be used and
orthogonal binding measurements must be made in order to interpret the rolling
parameter values.

*Reviewer #3:*

In this article the authors introduce a new experimental method, named mechanically
transduced immunosorbent (METRIS) assay, enabling the direct measurements of weak
interactions between proteins. Such protein-protein interactions (PPT) are typically
very weak (on the order of a fraction of the thermal energy, kBT). This new
technique is based on densely attaching one type of protein (antigen) to a
ferromagnetic micron bead, which can then bind to another protein, that has been
attached to a flat surface. By applying an external, rotating magnetic field, the
probe particles will be driven to perform a rolling motion on the flat surface. In
order to see any displacement of the probe particle the antigens need to detach on
one side and bind on the opposing side in the direction of the motion induced by the
magnetic field. This apparent friction informs us on the strength of the PPT. The
authors have tested a number of important PPT's – the non-covalent Interactions
between Ubiquitin-like Domains and Ube2D1 is one example.

An important strength of the new technique the authors introduce is the sensitivity
of their new method – it exceeds that of common methods by far, as they were able to
measure protein-protein interactions at much lower concentrations (2 orders of
magnitude) than is typically done. This allows a systematic determination of many
interaction potentials between the complex configurations of proteins as function of
solvent parameters and I guess also as function of temperature, which could be
interesting new the denaturation region. In addition, the affinity or attractive
interactions between specific proteins or RNA and proteins may change as function of
concentration: As the reaction constant or KD value depends directly on the total
concentration in the system the up- and down regulation processes of proteins may be
better understood. The authors demonstrate that they can achieve such a sensitivity,
which is well explained in the results and discussion.

The only point that needs to be addressed is the interpretation of the data. While
the systems and references are very well presented and researched it would be
helpful to the reader to explain in more detail the actual measurements and in
particular their interpretation and relation between the displacements measured and
the KD and Gibbs free energy differences.

My main suggestions are of technical nature concerning the interpretation of the
results and the underlying theory.

In the present description (line 1567) the authors relate the displacement of the
probe bead to a parameter they introduce as rolling parameter RP: zeta = δ x/(pi
times the diameter of the bead times the frequency of the oscillating magnetic
field). First I would call RP = zeta, or only RP unless I have not understood the
difference. However, my main questions is how did the authors derive this expression
in equation 1. It is clear that zeta must be dimensionless, hence it is presented as
displacement divided a velocity that is multiplied with the actuation time. This
velocity is the rolling velocity = angular velocity times the radius of the probe
colloid. It is confusing what π is doing there. The authors should double check the
equation and also explain to the reader why you introduce this parameter. Moreover,
the radius of the colloids sured here should be given.

If I understand right, this zeta or rolling parameter is at the same time normalised
against the streptavidin-biotin binding, which is known to be the strongest
non-covalent binding in the system. If the probe particle is brought in contact and
assuming that there is full coverage of avidin and biotin on the opposing surfaces,
the effective contact area will depend on the size of the particle, and the binding
strength with be more than 100 kBT. This means there is full traction and now slip
or friction between the probe particle and the support surface. But with such strong
binding energies, this means the fields applied must be very large and as the
particle roller over the surface the biotin or avidin on one of the surfaces must be
ripped off. Such behaviour was observed previously in AFM contact experiments, e.g.
by the group of Herman Gaup in the 90's. First, the authors need to give details
about the reproducibility of these reference measurements, in particular, the
applied Forces (e.g. Magnetic field) grafting density and effective number of bonds
in the contact area (which also may be influenced by the roughness of the colloidal
bead surface) need to be detailed. Secondly, the authors do not discuss or present
any calibration of the friction or rolling parameter. It is not necessarily a linear
function between zero (no interaction and thus no friction) and one (full
sticking/traction like when riding a bicycle).

In this context, it is important to look also into the displacement velocity. In
these measurements the friction relies on the fact whether the protein-protein
interactions are easily pealed off on one side and possibly re-established on the
front of the rolling motion. But this will depend on how fast the rolling motion is.
Again, AFM work by Gaup and in particular by Evan Evans (Annu. Rev. Biophys. Biomol.
Struct. 2001. 30:105-28 – which should be referenced) showed that the force or
interactions measured depend on how fast one pulls on the protein-protein link.
Hence, the authors should give much more experimental detail and relation to known
measurements. The numbers presented here for the Binding energies and their relation
to the rolling parameters must be clarified as well, so that it would be possible
for others to reproduce these measurements.

To summarise, the technique and the measured parameters need better elucidation and
validation to be useful to others. In Particular friction is not necessarily the
right terminology used in these measurements.

---

## [Author Response]

Reviewer #1:The authors are seeking to develop a method for measuring the binding affinities
between biomolecules using low amounts of material. The method described in this
manuscript uses a magnetic field to cause a ferromagnetic particle to roll along
a surface. When the particle is modified with one biomolecule and the surface is
modified with another, there is an increase in friction if the two molecules
interact. The degree of interaction is proportional to the amount of
friction.Strengths– The method requires a minimal amount of material.– The rolling parameter can be measured for many ferromagnetic particles
simultaneously, leading to robust statistical analyses.

We thank the reviewer for these comments and agree that these are some of the
practical strengths of this new technique, in addition to the high resolution,
simplicity, and ability to measure weak interactions using minimal amount of
material. Furthermore, by immobilizing both binding partners, the assay is not
dependent on mass action, and therefore, interactions of any strength can be
measured with a small quantity of protein.

Weaknesses– The method itself has already been described in a paper published by one of the
corresponding authors in 2014.

We appreciate the reviewer for pointing out this potential concern with the
manuscript. We understand that the previous wording of the manuscript, particularly
the following phrasing, might have accidently been misleading:

"Previous studies have demonstrated the utilized of the METRIS assay to measure
rolling parameters for a variety of interactions (e.g., Protein A-Fc,
histidine-metal, and Protein-PIP lipid interactions) [42-44]."

While the statement could be misinterpreted to mean that the assay was previously
developed, the PRL paper was actually focused on trying to achieve artificial
chemotactic directed motion based on differences in the effective friction
coefficient across a substrate. This was accomplished by creating a gradient in the
density of binding sites by passivating a biotin substrate with a droplet of avidin
and allowing it to evaporate and leave coffee-ring-like gradients in the density of
binding sites. This paper was a physics paper, and we weren't interested or aware of
the possibility of deducing binding affinity. It was only after this paper that we
recognized the potential of using the concept of effective friction to detect
differences in binding affinity by measuring translational displacement. There are
also several critical differences between the 2014 PRL paper and this manuscript.
The 2014 PRL paper utilized superparamagnetic walkers composed of two 3 μm diameter
polystyrene and iron oxide beads which align their magnetic moments in the presence
of a rotating magnetic field produced by two coils that were stochastically
actuated. We were not focused on measuring the biotin-streptavidin interaction, and
in fact, we could not measure the biotin-streptavidin interaction without heavy
passivation due to the limited magnetic torque that could be applied due to the
small magnetic moment of our superparamagnetic particles. The current technique
utilizes a single roller 10μms in diameter, a glass substrate functionalized with
avidin, and a 3D Helmholtz coil like apparatus to drive walker motion. Additionally,
this current study is exclusively focused on correlating differences in the
translational displacement (via introduction of a rolling parameter) to binding
affinity and extrapolating an estimated K_d_ for PPIs. Given these
significant differences, we have revised the manuscript to indicate that the
foundations for this technique to be applied were previously published. The other
citations in this section were to a thesis and a patent, which are not peer reviewed
publications, which have some of the experimental optimization, but were not able to
measure protein-protein affinities. However, this previously cited works were
investigating a primarily physics phenomenon, whereas here we have further developed
this technique to measure differences in binding affinity.

We would like to thank the reviewer for pointing out this potential area of
confusion, and we have significantly changed the manuscript, spending more time
discussing the experimental results of the streptavidin-avidin streptavidin-biotin
experiment in Figure 1, since this is the first reporting of that experiment.
Overall, this new format really improves the readability and clarity of the
manuscript.

– Insufficient detail is provided on the method and output. Raw data showing that
the rolling particle is displaced in the presence of a binding partner on the
surface is missing.

We appreciate this comment and agree that being able to visualize and work with the
raw data would be very helpful to understand the technique and potential
applications. To that end, we have made significant changes to the manuscript to
illustrate more raw data in the presence of non-binding, extremely strong binding
(biotin-streptavidin), and other binding PPI partners with differing affinity.

1. We have made raw videos from some of the conditions we tested available as a
supplemental video. These raw videos should be very helpful in visualizing the
differences in the mechanical signal observed, translational displacement due to the
presence of the binding interaction.

2. For figure 2,3,4, we have included a panel which shows the average distance that a
roller translates per actuation.

3. We have also included some animations that illustrate the translational
displacement of the beads, in micrometers, for the different interactions to
illustrate even small differences in binding affinity can be deduced visually by the
differences in the translational displacement of the rollers.

4. We have included histograms of the rolling parameters of the beads for all
interactions, and these show clear differences. As seen for all the interactions
(shown in Author response image 1), the
distributions are Gaussian, but we can see some stark differences in the width of
the interactions, particularly the PPI interactions.

**Author response image 1. respfig1:** Distribution of rolling parameters for different PPI
interactions.

5. We have substantially revised figure 1 to include clockwise and counter-clockwise
actuation for Streptavidin-Biotin and Streptavidin-Avidin (Figure 1C), More detail
about the roller trajectory (Figure 1D), and included the rolling parameter
distribution

We believe this additional raw data will help clarify how this technique's mechanical
signal output is utilized to differentiate between binding affinities, and we hope
that these videos illustrate the novel nature and ease of use of this technique. If
there are any other raw data the reviewers think should be included in the
manuscript, we are happy to include it.

– The Kd cannot be directly obtained from the rolling parameter.

While it is true that the K_d_ cannot be directly obtained in isolation at
this time, this fact does not take away from the importance of this technique and
possible implications. With further work and more data, it will be possible in the
future to derive a mathematical model to calculate K_d_ directly or develop
a fundamental theoretical model that will enable one to directly measure binding
affinity from the rolling parameter value. Nonetheless, this is the first report of
a technique that can be utilized to deduce differences in binding affinity from a
mechanical signal, the translational displacement. Also, even without a
K_d_ based on a standard curve, related molecules' binding strengths
can still be measured with reasonable accuracy using the ΔΔG method as described in
the manuscript.

– Unlike methods such as surface plasmon resonance and biolayer interferometry,
which use a comparably low amount of material, this method requires both binding
partners to be affinity-tagged. The tags and the immobilization could obscure
native interactions.

We agree that METRIS requires both proteins be immobilized with high-affinity tags,
which could obstruct some biological interactions. However, this issue is not unique
to METRIS. Both SPR and BLI have one protein immobilized, usually through an
affinity tag (not necessarily a high-affinity tag) or surface adsorption, which can
also block interactions. In this study, we present two methods of tagging the
protein at different locations, through a BirA tag and with maleamide-biotin, which
in this study showed comparable RPs, (Supplemental Figure 3) so this gives the user
flexibility in how to achieve immobilization, and there are many other possible
immobilization strategies that we are testing. Importantly, this also indicates that
we can see similar results from both approaches. We also have added some additional
text in the Discussion section of the manuscript about this. For the interactions we
measured in our study, we have no indication that tagging blocked any of the
interactions, given the strong correlation of our data to other studies that
measured these protein-protein interactions.

Additionally, immobilizing both proteins affords advantages to SPR and BLI because we
use significantly less material than either BLI and SPR. However, in BLI and SPR,
the soluble analyte is still dependent on mass action and thus can require high
concentrations of protein to measure weaker affinities, still consuming a lot of
materials and also leading to potential thermodynamic non-ideality of the soluble
analyte. We have a more detailed comparison between SPR/BLI and METRIS below and
directly compare BLI and METRIS for low-affinity binders.

– Biomolecular interactions have a wide range of affinities that span over 15
orders of magnitude (1 M to 10-15 M). Yet, the dynamic range of the rolling
parameter is only one order of magnitude (0.081 for Kd of 1 M and 0.918 for Kd
of 10-15M). As a result, the rolling parameter values cluster together when
correlated with Kds on a logarithmic scale. Subtle differences in binding
affinity can therefore be lost.

We believe that this is one of the key properties that makes the METRIS technique
valuable as a bioassay platform, and can clarify this point. While the rolling
parameter only spans one order of magnitude, our manuscript and other preliminary
data illustrate that METRIS can measure affinities spanning 15 orders of magnitude,
but is also sensitive enough to measure differences as low as 2-fold like we
observed when measuring the Orc1-BAH interaction with methylated peptides (See
Figure 3). Thus, we do not believe we are losing subtle differences in binding. This
accuracy is due to the very small uncertainty or error that we report and this may
be because of some confusion with the definition of the rolling parameter, which we
have clarified in the manuscript in response to one of the comments by Reviewer #3
(see below). The rolling parameter is the ratio of the observed, experimental
translational displacement of the roller, Δx, to the maximum theoretical
displacement of a rolling sphere. The maximum displacement of a rolling sphere that
hinges perfectly on a single point will simply be the circumference of the sphere
(πD, where D is the diameter of the sphere typically 10 μm for our rollers, but we
explicitly measure the diameter for every roller) times the number of rotations
which is governed by the rotational frequency of the magnetic field, ω, which is 1Hz
for this manuscript, times the actuation period in seconds, τ, which is 5s for this
experiment. Thus, the maximum theoretical displacement is simply πDωτ. A rolling
parameter of 1 would equal 157 µm, where a rolling parameter of 0.082, the null
binding measurement in this manuscript, would correspond to a distance of 13 µm. We
can measure the translational displacement and the diameter of the particle with
high accuracy using a custom-written Mathematica analysis script in conjunction with
Able Particle tracker, and there is virtually no uncertainty for the actuation
period and the rotational frequency of the magnetic field in this technique. The
recorded images have a resolution of approximately 1μm per pixel, and our image
analysis software has subpixel resolution so we can measure subtle differences in
displacement on the order of 100s of nanometers. Couple this with the fact that we
measure on average 20-30 rollers. Each roller rolls approximately 36 times, 5
seconds each roll, meaning we roll the entire sphere 5 complete rotations, we have
incredible statistical averaging. This METRIS technique and probes are extremely
sensitive and can distinguish forces on the order of 10s of femtonewtons, as
previously reported (PRL 2014), and we have added this to the main text of the
manuscript.

This is also somewhat akin to thermodynamics, while biological interactions span 15
orders of magnitude, the ΔG/mol between the bound and unbound states only span ~2
orders of magnitude (~20 kcal/mol). Moreover, we would like to highlight that for
these interactions, which are very weak and can be impossible to measure with other
traditional techniques like BLI, METRIS is indeed sensitive enough to distinguish
and measure a wide range of binding affinities with high resolution.

– For any given system, the Kd must be measured using an orthogonal technique in
order to establish binding affinities that can be correlated with the measured
rolling parameter. More specifically, the rolling parameter values are not
generalizable from system to system.

We agree with the reviewer's comment. As we acknowledge in the text, this is one of
the limitations of METRIS at this point, as we do not have a robust theoretical
model that allows for the direct calculation of binding affinity from the rolling
parameter. We are continuing to work on this aspect of the project. Still, at this
point, even an empirical and orthogonal calculation of binding affinities using
METRIS is valuable at this point for the following reasons:

(1) This technique does not utilize a large amount of material, so even at this
stage, METRIS can be very useful as a quick diagnostic technique for binding
partners that are difficult to produce in large quantities.

(2) METRIS is relatively simple to perform, and one can quickly obtain at the very
least a qualitative comparison between several different binders within a family of
binders/interactions.

(3) METRIS is well suited to measure very weak interactions, and as we demonstrated
in this paper – even when studying characterized systems, we can still produce
biological insights into interactions. Furthermore, even without an empirically
determined K_d_ using an orthogonal system, you can still derive relatively
accurate ΔΔG using the standard curve from our aggregate data for a series of family
members.

(4) If a user establishes a baseline, the results can be highly quantitative compared
to other methods for the reasons described.

(5) METRIS can still be very useful even as a qualitative bioassay screener,
especially as we improve throughput.

Additionally, similar calibration curves are required to get quantitative results
from ELISA measurements; however, our sensitivity and reproducibility is well beyond
the ability of any quantitative ELISA with significantly less material utilized.

– The schemes presented in Figure 1 are difficult to interpret. Perhaps it would
be better show a video of the particles rolling along a surface with no binding
partner and another video when there is a binding partner. In the absence of
these raw data for each system, the binding measurements are not compelling.

We really appreciate this comment from the reviewer, and we have made the recommended
changes in Figure 1 and included more supplemental videos and supplemental
figures.

– The authors spend a great deal of time comparing their method to NMR, why?
Other measurements such as SPR and BLI would be more appropriate
comparisons.

In the manuscript, we focused on NMR because it is the standard for measuring very
weak interactions. However, the reviewer is correct that SPR and BLI can use small
quantities of materials and may be more similar to METRIS, given that one of the
interacting partners is immobilized. We have added some text to the manuscript to
address this point.

Additionally, METRIS has significant advantages to SPR and BLI, especially when
examining very weak binders. Both SPR and BLI are dependent on mass action, so the
soluble analyte still must be above the Kd, and for weak binding, this still
requires a significant amount of protein. For example, in another unpublished study
where we used METRIS in combination with BLI to measure weak binding to ubiquitin
for three mutants, we needed 5.5 mLs of ubiquitin at 5mM (~200 mg) to measure
affinities in the 10^-4^-10^-3^ range (see Author response image 2). For very weak binders, it is not
possible to measure the K_d_. When fitting BLI data, it is critical to
establish the endpoint, i.e., saturate the binding curve, and one often needs very
high concentrations of proteins. With BLI, we could not accurately measure the
K_d_ values for the weakest binders (compare results from METRIS and
BLI for mutants 2 and 3 in Author response image 2) because we did not saturate the binding curves, and at higher
ubiquitin concentrations, thermodynamic non-ideality occurs (Author response image 2 and D). With METRIS, we could
accurately measure these affinities with 30 pmols of protein total, and we could
even measure the affinity for mutant 3, which we couldn't fit with BLI (Author response image 2 A-C). For these BLI
measurements, we needed 23 µmols of material – ~100,000 times more material than we
needed for METRIS. For most proteins, it is impossible to obtain that much protein
at that high of a concentration. Moreover, the BLI signal is noisy and is dependent
on accurately measuring the ubiquitin concentration (Figure 2D).

Additionally, for BLI and SPR, it is difficult to distinguish between background
binding to the chip when studying weak interactions, where for METRIS, we have good
sensitivity between the null and weak binding range.

**Author response image 2. respfig2:** Comparison Between BLI and METRIS. (A) Rolling parameter measurements for 3 mutants that bind ubiquitin. (B) Fit
of METRIS to Kd values. (C) Kdvalues obtained from METRIS and BLI. (D) BLI
data for each of the mutants.

For SPR, sometimes you can derive binding values below the K_d_, but in the
scenarios, only a fraction of the chip is bound, and depending on your signal, you
may not be able to detect binding. Also, the signal for BLI and SPR is dependent on
a significant size change (i.e., change the refractive index), so for some binding
experiments, like a peptide/small molecule to a protein, the signal can be very
small. Given these issues, we think that METRIS has significant advantages to BLI
and SPR. We have incorporated this discussion into the manuscript and highlighted
the utility of METRIS as a new technique.

– In general, I found the way in which the manuscript was written to be very
misleading. After reading the Intro, I was expecting the rest of the paper would
focus on the development of the method. Then, in lines 184-185 on page 6, it
becomes clear that the method was already published 7 years ago by one of the
corresponding authors. Thus, the paper is less about the method and more about
the application.

We appreciate the reviewer's comment, and this is due to some confusion about how the
manuscript was written. Above, we described the key differences between the 2014 PRL
paper, and we have provided more background about how this method is different from
previously described techniques. We have also made changes to the manuscript to
avoid this potential source of confusion.

– Please remove sentences like "It is difficult to overstate how
transformative this will be for the study of PPIs." The method is not a
direct measurement of binding affinity. The dynamic range of the measurement
does not correlate with the range of biomolecular binding affinities. Multiple
affinity tags must be used and orthogonal binding measurements must be made in
order to interpret the rolling parameter values.

We have removed the claims. We were attempting to highlight the importance of the
METRIS technique but appreciate this comment and have made subsequent changes to the
manuscript to ensure the claims made are reasonable, accurate, and not
overstated

Reviewer #3:In this article the authors introduce a new experimental method, named
mechanically transduced immunosorbent (METRIS) assay, enabling the direct
measurements of weak interactions between proteins. Such protein-protein
interactions (PPT) are typically very weak (on the order of a fraction of the
thermal energy, kBT). This new technique is based on densely attaching one type
of protein (antigen) to a ferromagnetic micron bead, which can then bind to
another protein, that has been attached to a flat surface. By applying an
external, rotating magnetic field, the probe particles will be driven to perform
a rolling motion on the flat surface. In order to see any displacement of the
probe particle the antigens need to detach on one side and bind on the opposing
side in the direction of the motion induced by the magnetic field. This apparent
friction informs us on the strength of the PPT. The authors have tested a number
of important PPT's – the non-covalent Interactions between Ubiquitin-like
Domains and Ube2D1 is one example.An important strength of the new technique the authors introduce is the
sensitivity of their new method – it exceeds that of common methods by far, as
they were able to measure protein-protein interactions at much lower
concentrations (2 orders of magnitude) than is typically done. This allows a
systematic determination of many interaction potentials between the complex
configurations of proteins as function of solvent parameters and I guess also as
function of temperature, which could be interesting new the denaturation region.
In addition, the affinity or attractive interactions between specific proteins
or RNA and proteins may change as function of concentration: As the reaction
constant or KD value depends directly on the total concentration in the system
the up- and down regulation processes of proteins may be better understood. The
authors demonstrate that they can achieve such a sensitivity, which is well
explained in the results and discussion.

We truly appreciate the comments made by the reviewer and appreciate the insight into
the METRIS technique. We agree that METRIS will allow scientists to study
interactions in different ways than other methods and we are excited to test some of
the experiments that the reviewer mentions above.

The only point that needs to be addressed is the interpretation of the data.
While the systems and references are very well presented and researched it would
be helpful to the reader to explain in more detail the actual measurements and
in particular their interpretation and relation between the displacements
measured and the KD and Gibbs free energy differences.My main suggestions are of technical nature concerning the interpretation of the
results and the underlying theory.In the present description (line 1567) the authors relate the displacement of the
probe bead to a parameter they introduce as rolling parameter RP: zeta = δ x/(pi
times the diameter of the bead times the frequency of the oscillating magnetic
field). First I would call RP = zeta, or only RP unless I have not understood
the difference. However, my main questions is how did the authors derive this
expression in equation 1. It is clear that zeta must be dimensionless, hence it
is presented as displacement divided a velocity that is multiplied with the
actuation time. This velocity is the rolling velocity = angular velocity times
the radius of the probe colloid. It is confusing what π is doing there. The
authors should double check the equation and also explain to the reader why you
introduce this parameter. Moreover, the radius of the colloids sured here should
be given.

This is a great comment. We have made changes in the text that correct the
description of the rolling parameter, which initially and mistakenly stated
normalization by the theoretical velocity, which was not correct. We have also
changed the labeling of the zeta to the RP. The RP is the ratio of the actual
translational displacement of the roller, Δx, divided by the maximum theoretical
displacement of a rolling sphere on a substrate. One can imagine that the maximum
distance a sphere can roll on a substrate would be to hinge on a single point and
then roll perfectly on this point, whereby the sphere would translate a distance
equivalent to the sphere's circumference. In this scenario, our sphere or roller
undergoes 5 complete rolls per actuation period as we kept the rotational frequency
of the magnetic field constant at 1Hz and the time of rolling was 5 seconds, so the
roller undergoes 5 complete rotations per roll. So, the maximum theoretical
translational displacement of a sphere would be the circumference of our sphere π D,
multiplied by the actuation time τ, and the rotational frequency of the magnetic
field ω. We introduce this parameter because, as the reviewer points out, the
translational displacement of the roller depends on several factors beyond that of
binding partners, and those include D, τ, and ω. We introduced the rolling parameter
given this dimensionless number allows us to provide the reader with an
apples-to-apples comparison of how the translational displacement varies across each
PPI. Additionally, and pragmatically, although our particle diameters are nominally
approximately 10 μm, they tend to vary slightly, but these differences affect the
rolling parameter (which is why we explicitly measure each roller diameter) and thus
the extrapolated binding affinity. The introduction and creation of the rolling
parameter is essential to provide an accurate comparison of the effective friction
induced by binding and, therefore, the translational displacement of the rollers.
Without such a parameter, the extrapolation to a binding affinity would be
impossible or, at the very least, much less robust.

If I understand right, this zeta or rolling parameter is at the same time
normalised against the streptavidin-biotin binding, which is known to be the
strongest non-covalent binding in the system. If the probe particle is brought
in contact and assuming that there is full coverage of avidin and biotin on the
opposing surfaces, the effective contact area will depend on the size of the
particle, and the binding strength with be more than 100 kBT. This means there
is full traction and now slip or friction between the probe particle and the
support surface. But with such strong binding energies, this means the fields
applied must be very large and as the particle roller over the surface the
biotin or avidin on one of the surfaces must be ripped off. Such behaviour was
observed previously in AFM contact experiments, e.g. by the group of Herman Gaup
in the 90's. First, the authors need to give details about the reproducibility
of these reference measurements, in particular, the applied Forces (e.g.
Magnetic field) grafting density and effective number of bonds in the contact
area (which also may be influenced by the roughness of the colloidal bead
surface) need to be detailed. Secondly, the authors do not discuss or present
any calibration of the friction or rolling parameter. It is not necessarily a
linear function between zero (no interaction and thus no friction) and one (full
sticking/traction like when riding a bicycle).

We appreciate the reviewer’s keen insight. Indeed, we calibrate our METRIS platform
by looking at the RP of a non-interacting binding pair and that of the strongest
non-covalent interaction biotin and streptavidin. As we mention in the manuscript,
we fully coat our probe and substrate with more than 50X the theoretical binding
capacity to ensure a fully coated substrate and particle. We do this to ensure that
the density of binders remains as constant as possible, but as the diameter of the
particles is slightly different, the number of binders will change slightly, hence
why we also introduce the rolling parameter to account for these differences. The
density of biotin-binding sites per roller is on the order of 6 x 10^8^
biotin molecules per roller, which corresponds to a density of biotin-binding sites
on the order of 5 x 10^10^ binding sites per mm^2^, which is of
similar density to the density of streptavidin binding sites is reported in the
manuscript. The actual footprint of the sphere on the substrate is not a trivial
calculation, but assuming that footprint is 1% of the projected area, that would
result in approximately 10^5^ potential binding sites, but due to the size
of tagged PPI, this number of binding interactions is likely less. Regardless to
break these interactions will require an applied magnetic torque, τ, which is
defined as the m x B where m is the magnetic moment, and B is the applied magnetic
field strength. As described in the manuscript, the applied magnetic field strength
is approximately 10mT, and the magnetic moment is on the order of 10^-11^ A
M^-2^ this would produce a torque large enough to break the
biotin-streptavidin bond. In our previous study in 2014, we studied a system
composed of walking streptavidin-coated superparamagnetic dimers, 3 μm in diameter
each, the biotin-streptavidin bond was not able to be broken without heavily
passivating the biotin substrate with avidin particles and coating the walkers with
biotin-PEG. This combination reduced the number of biotin-binding sites, and due to
the PEG brush, the biotin-streptavidin interaction strength was reduced because
compressing the polymer PEG brush is entropically unfavorable. The superparamagnetic
particles were unable to break the biotin-streptavidin interaction even when using
approximately the same magnetic field strength because the magnetic moment of the
walkers is more than an order of magnitude less than that of the ferromagnetic
rollers. Thus, the torque is more than an order of magnitude less.

However, despite this increase in the magnetic torque, we are not stripping the
biotin from the substrate in this system. This is simply not possible in the system
that we are using because the biotin or the streptavidin is covalently linked to the
glass substrate, and we are not breaking covalent bonds, just breaking the
biotin-streptavidin interaction. If the biotin or avidin was being ripped off from
the substrate and remained stuck on the roller, there is no way for these ligands to
re-attach to the substrate. Thus, if this is indeed the mechanism when the rollers
are actuated in the reverse direction, we should see a drastic decrease in the
rolling parameter as one of the binding partners has been removed. However, this is
not observed, as seen in Author response image 3 and in our supplemental movies. This is distinct from scenario where
binding partners are being ripped from the surface. Take for example the data from a
pre-print of a system consisting of a supported lipid bilayer that has biotinylated
lipid bilayers. The supported lipid bilayer is not covalently attached to the
substrate and in here the binding affinity of the biotin-streptavidin interaction is
enough to rip the lipids from the substrate and you can see in Author response image 3 as the lipids are ripped from the
bilayer the rolling parameter decreases as the roller continues to roll, thus
illustrating that in this system in the manuscript we are not ripping off binding
partners from the substrate.

**Author response image 3. respfig3:** Rolling parameter of biotin-streptavidin interaction on avidin substrate
(A) and a biotinylated supported lipid bilayer (B).

Finally, in terms of calibrating the RP to friction, we clarify that the RP gives us
an indication of the effective friction induced by binding interactions, and we
correlate RP to the binding affinity. We appreciate the reviewer for this comment
and have updated the manuscript to make this distinction more clear. We do not
assume that the RP and friction scale linearly and have changed the manuscript. We
also want to thank the reviewer for the comments and hope that these experimental
details the distributions of the rolling parameter values given in Figure 1
illustrate the robust and reproducible nature of the technique.

In this context, it is important to look also into the displacement velocity. In
these measurements the friction relies on the fact whether the protein-protein
interactions are easily pealed off on one side and possibly re-established on
the front of the rolling motion. But this will depend on how fast the rolling
motion is. Again, AFM work by Gaup and in particular by Evan Evans (Annu. Rev.
Biophys. Biomol. Struct. 2001. 30:105-28 – which should be referenced) showed
that the force or interactions measured depend on how fast one pulls on the
protein-protein link. Hence, the authors should give much more experimental
detail and relation to known measurements. The numbers presented here for the
Binding energies and their relation to the rolling parameters must be clarified
as well, so that it would be possible for others to reproduce these
measurements.

We thank the reviewer for pointing us to the excellent manuscript by Evan Evans and
also pointing us to relevant papers from the field of force spectroscopy. We have
added these citations to the manuscript and have also added a section in the
discussion about force spectroscopy and compared it to METRIS. We would also like to
point the reviewer to the previous comments regarding the interactions being peeled
off the substrate. In terms of the protein-protein interactions begin broken, as the
roller rolls, this does indeed depend on the rotational frequency of the magnetic
field. In this study, the rotational frequency is kept fixed at 1Hz as this has been
determined as an appropriately slow rotational frequency to allow for binding
events. If one increases the magnetic field's rotational frequency, there will be
less time for binding, and the effective friction will decrease, as you can see in
Author response image 4. Here we find
that at 5Hz, most interactions are no longer measurable as the bead is rotating too
fast, and binding cannot occur because there is not sufficient time for binding. In
fact, only the biotin-streptavidin interaction is observed at 10Hz, but as you can
see, the rolling parameter has decreased drastically. We hope this new information
makes it easier for comparisons.

**Author response image 4. respfig4:** Rolling parameter as a function of rotational frequency for
interactions.

To summarise, the technique and the measured parameters need better elucidation
and validation to be useful to others. In Particular friction is not necessarily
the right terminology used in these measurements.

We have made the appropriate changes to the manuscript and included the information
suggested above. We have also changed the term friction to effective friction
induced by binding to be more accurate. However, the reviewer is correct that the
particle is essentially slipping along and what we are measuring is how effectively
the rotational torque is converted to rotational motion. Thus, we have simplified
this concept and describe it as effective friction. Again, we thank the reviewer for
these comments, which have improved the quality of the manuscript.